# Improving land-atmosphere coupling in a seasonal forecast system by implementing a multi-layer snow scheme

Eunkyo Seo[1,2], Paul A. Dirmeyer[2]

[1] Department of Environmental Atmospheric Sciences, Pukyong National University, Busan, 48513, Republic of Korea
[2] Center for Ocean-Land-Atmosphere Studies, George Mason University, Fairfax, Virginia, 22030, United States

*Correspondence to*: Eunkyo Seo (eseo@pknu.ac.kr)

**Abstract.** This study explores the influence of implementing a multi-layer snow scheme on the climatological bias within a seasonal forecast system. A single layer snow scheme in land surface models often inadequately represents the insulating effect of snowpack, resulting in warm and cold biases during winter and snow melting seasons, respectively. By contrast, multi-layer snow schemes enhance energy transport between the land and the atmosphere. To investigate this impact, two versions of the Global Seasonal Forecast System (GloSea) – GloSea5 with a single layer snow scheme and GloSea6 with a multi-layer snow scheme – are compared over 24 years (1993–2016). Results shed light on the significance of accurately representing the insulating effect of snow in improving retrospective seasonal forecasts. In GloSea6, the snow melting season shifts two weeks later, delaying the onset of evaporation in the spring season. This slows soil moisture drying, resulting in an improvement in its climatology and memory. The abundant soil moisture enhances the partitioning of incoming energy into latent heat flux, allowing for more evaporative cooling at the surface, and constrains water-limited coupling. Such improvements in the land surface processes, especially over the mid-latitudes, mitigate the near-surface warming bias over the entire diurnal period and the oversensitivity of atmospheric conditions to the land surface variability. The model performance in simulating precipitation is also improved with the increase in precipitation occurrence over snow-covered regions, significantly reducing model error in the Great Plains, Europe, and South and East Asia. Above all, this study demonstrates the value of implementing a multi-layer snowpack scheme in seasonal forecast models, not only during the snowmelt season but also for the subsequent summer season, for model fidelity in simulating temperature and precipitation along with the reality of land-atmosphere interactions.

## 1 Introduction

Subseasonal-to-seasonal (S2S) forecasts have become increasingly pivotal in numerous fields, encompassing agriculture, water resource management, energy, transportation, and disaster preparedness. The significance of S2S forecasting stems from their ability to provide actionable insights into forthcoming weather and climate conditions over the span of weeks to months. The predictability of S2S forecasts is strongly tied to the quality of the initial conditions and data assimilation technique, which mathematically finds optimal values with minimized analysis errors to merge observations into a dynamical model, has been employed to create improved global analyses (Seo et al., 2021; Kumar et al., 2022). Forecasts across various time scales

underscore the necessity for precise initial states of distinct components within the forecast model, as each component retains information over inherently disparate time scales (Richter et al., 2024). As the memory of initial land conditions can extend out to approximately 2 months, the importance of realistic land surface initialization in determining skill of the subseasonal forecast is paramount (Koster et al., 2011; Guo et al., 2011; Seo et al., 2019).

In particular, soil moisture (SM) plays a pivotal role in hydrological and meteorological dynamics, acknowledged as an essential climate variable by the World Meteorological Organization (WMO) (Seneviratne et al., 2010; Santanello et al., 2018). Its persistence or memory can significantly enhance forecast accuracy, particularly at time scales extending to 1–2 months (Dirmeyer et al., 2016; Dirmeyer et al., 2018; Seo and Dirmeyer, 2022b). The fidelity of modelled SM contributes to a more accurate portrayal of land-atmosphere interactions, facilitating the exchange of water and energy fluxes at the land surface (Seo et al., 2024). This enhanced representation holds potential for predicting extreme climate events, particularly those intensified by land-atmosphere feedbacks within extended range forecast systems (Seo et al., 2020; Dirmeyer et al., 2021; Tak et al., 2024). SM is directly constrained by the components of the typical water balance equation: precipitation, latent heat flux, and runoff, but the modelled snow affects the representation of snow characteristics.

The pivotal role of snow in land-atmosphere interactions highlights the significance of accurately representing cold processes related to snow in hydrometeorology and dynamical predictions. Compared to other land surface variables, snow exhibits distinctive characteristics such as high albedo, high thermal emissivity, and low thermal conductivity, which profoundly influence radiation budget and surface moisture and energy fluxes to the atmosphere. The presence or absence of snow can result in a disparity of approximately 10 K in the climatology of surface air temperature (Betts et al., 2014). This discrepancy primarily stems from the reduction in net shortwave radiation attributable to the high albedo of snow. Snow-atmosphere feedback is evolved in three distinct stages: before, during, and after snowmelt. Meanwhile, the coupling strength is strongest during snowmelt and the coupling strength after snowmelt (delayed soil moisture impact) is stronger than that before snowmelt (radiative impact from surface albedo) (Xu and Dirmeyer, 2011). Therefore, during the warm season, SM dynamics are intricately linked to the physical characteristics of snow, affecting the initiation of evaporation due to snowmelt. It plays a crucial role in determining the model's ability to accurately simulate atmospheric variables through land-atmosphere coupling processes.

Land surface models (LSMs) have not often utilized a multi-layer snowpack scheme, which has proven insufficient in accurately capturing the seasonal evolution of snow cover. Consequently, this approach tends to result in warm and cold biases during winter and snow melting seasons, respectively. Addressing these limitations, recent advancements in LSMs aim to integrate a multi-layer snow scheme to enhance the representation of snow dynamics and mitigate associated biases. For instance, Noah-MP represents the latest iteration of Noah LSM which is a land component in many regional and global operational forecast models, featuring numerous enhancements to improve the realism of biophysical and hydrological processes (Niu et al., 2011). Notably, for a more accurate representation of snow physics, Noah-MP integrates a layered snowpack scheme. This scheme dynamically adjusts the number of snow layers based on the depth of snow, ensuring a more realistic conceptualization of snow accumulation and melt processes. The Joint UK Land Environment Simulator (JULES)

Land Surface Model (LSM) features the utilization of a multi-layer snow scheme in its current operational system. This
implementation also demonstrates enhancements in the representation of land surface processes (Walters et al., 2017). JULES
is incorporated within the GloSea forecast system (Maclachlan et al., 2015).

Numerous studies have aimed to improve the sophistication of snow physics and highlighted its advancement in numerical
models (Xue et al., 2003; Arduini et al., 2019; Cristea et al., 2022). The impact of multi-layer schemes on S2S forecasts remains
inadequately explored and understood, even though all but three of 13 S2S models (BoM: POAMA P24, CNR-ISAC: GLOBO,
and NCEP: CFSv2) now use multi-layer snow schemes. Hence, this study conducts a comparative analysis between GloSea5
(single layer snowpack) and GloSea6 (multi-layer snowpack), past and present operational forecast systems at the UK Met
Office and the Korea Meteorological Administration (KMA), in retrospective forecasting in order to investigate the impact of
an advanced snow scheme. The primary objective of this study is to assess the seasonal cycle of snow and land surface variables
throughout the snow-covered period. Furthermore, this study assesses the model's capability to replicate the mean climatology
of key land surface and near-surface variables, e.g., surface SM, surface air temperature, and precipitation, during boreal warm
season. Daily mean, maximum, and minimum temperatures are validated at subdaily time scales to elucidate the time of
significant improvements in model performance. The model fidelity in the simulation of land-atmosphere interactions,
corresponding to water- and energy-limited processes, is also diagnosed to identify the realism of land coupling regime.

The paper is organized as follows. Section 2 describes the GloSea5 and GloSea6 models, and the validation datasets used in
this study. Section 3 provides the methodology to evaluate the model performance. Section 4 presents and discusses the results
of this study. Finally, Section 5 summarizes the results and their implications for future applications.

## 2 Data

### 2.1 Forecast Model

This study explores the performance of the Global Seasonal forecast system (GloSea) version 5 and 6, which are abbreviated
as GloSea5 and GloSea6, respectively. These are the fully coupled ensemble forecast models with atmosphere-land-ocean-sea
ice components, being developed by the UK Met Office. GloSea5 (Maclachlan et al., 2015) Global Coupled model 2.0 (GC2;
Williams et al., 2015) configuration consist of UM (Unified Model) version 8.6 atmospheric component (GA6.0; Walters et
al., 2017) having N216 horizontal resolution of 0.56° latitude × 0.83° longitude with vertically 85 hybrid-sigma coordinates
topped at 85 km, JULES (Joint UK Land Environment Simulator) version 4.7 land surface model (GL6.0; Best et al., 2011)
with four soil layers (0–10-, 10–35-, 35–100-, and 100–300-cm depth), as well as NEMO (Nucleus for European Modelling
of the Ocean) version 3.4 oceanic component (GO5.0; Megann et al., 2014) and CICE (Los Alamos Sea-ice Model) version
4.1 sea-ice component (GSI6.0; Rae et al., 2015) on an ORCA tripolar 0.25° global grid with 75 vertical levels. Those
components exchange interactive variables with the OASIS3 coupler (Valcke, 2013). GloSea6 Global Coupled model 3.2
(GC3.2) updates the atmospheric, land, ocean, and sea-ice components to the version of UM vn11.5 (GA7.2), JULES vn5.6

(GL8.0; Wiltshire et al., 2020), NEMO vn3.6 (GO6.0; Storkey et al., 2018), and CICE vn5.1.2 (GSI8.1; Ridley et al., 2018) without any modification in the resolution. The model components of GloSea6 are coupled with the OASIS3-MCT (Model Coupling Toolkit; Craig et al., 2017). We refer GloSea5 GC2 and GloSea6 GC3.2 to GloSea5 and GloSea6, respectively, throughout this paper.

Substantive changes in the GloSea6 compared with GloSea5, mostly in model physics, have been implemented throughout all model components (Kim et al., 2021). For instance, the atmospheric physics are modified in radiation (improving gaseous absorption through upgrades in McICA (Monte Carlo Independent Column Approximation) and parameterization in ice optical properties), microphysics (updates in warm rain parameterization and newly implementing ice particle size-dependent parameterization), cloud physics (including radiative effects from convective cores), gravity wave drag (implement heating

from gravity-wave dissipation), boundary layer (correcting cloud top entrainment during decoupling to the land), cumulus parameterization (improving updraught numeric in convective process and updating CAPE closure as a function of large-scale vertical velocity), and new modal aerosol scheme (UKCA GLOMAP (Global Model of Aerosol Processes) scheme; Mann et al., 2010). Aforementioned atmospheric physics updates in the GloSea6 are likely to improve the performance of model systemic errors, particularly in the overestimated vertical profile of cloud fraction at upper troposphere, tropospheric cold and

dry biases, the underestimated jet stream, the overestimated precipitation, and the negative bias of troposphere geopotential height during boreal summer (Williams et al., 2018).

In addition, there are two major updates in land physics: the implementation of a multi-layer snow scheme and the realization of shortwave surface albedo with wavelength dependence. GloSea5 has a single layer snow scheme, in which snow is assigned a constant thermal conductivity and density, allowing direct heat exchange between the surface atmosphere and the soil. It

combines the snow and the uppermost soil layer into a single thermal store, with the increased layer thickness accounting for the reduced thermal conductivity of snow. However, this scheme lacks proper closure of the surface energy budget (SF. 1) and a dynamic representation of snowpack evolution with the inadequate depiction of the snowpack's insulating properties. The improvement from the implementation of the multi-layer snow scheme is shown not only in the realization of the snow melt season, but also in the soil temperature and permafrost extent (Walters et al., 2017). For instance, the multi-layer snow scheme

leads to surface warming of the soil temperature during the winter season, as the heat flux from the soil to the atmosphere is reduced, but shows a surface cooling in the spring season, as the increase in insulating radiation inhibits snowmelt. In the snow frontal regions, the increase in land surface albedo is due to the delay in the onset of snowmelt by the multi-layer snowpack, while the decrease in surface albedo over the Sahara, the Arabian Peninsula, and India is related to the modification in land surface albedo physics as a function of shortwave wavelength. Other land surface physics are consistent in GloSea5 and

GloSea6. For land surface types, five vegetation (broadleaf trees, needleleaf trees, C3 grasses, C4 grasses and shrubs) and four non-vegetated surfaces (urban, open water, bare soil and permanent land ice) are classified and the monthly climatology of leaf area index, derived from MODIS satellite product (Yang et al., 2006), is prescribed corresponding to the plant functional types. Snow is present on every land tile, including inland water when its temperature is below freezing. Therefore, the climate

sensitivity over mid-latitude snow frontal regions is attributable to the implementation of the multi-layer snow scheme in the
GloSea6.

In terms of initial conditions for each model component, GloSea5 and GloSea6 commonly utilize ERA-interim and a variational data assimilation system for the NEMO ocean model (NEMOVAR; Mogensen et al., 2012) analysis for the atmospheric and ocean and sea-ice initializations, respectively. Land surface reanalysis, where the land offline simulation is forced by atmospheric boundary conditions from Japanese 55 years Reanalysis (JRA-55; Kobayashi et al., 2015) and European Centre for Medium-Range Weather Forecasts (ECMWF) Reanalysis version 5 (ERA5; Hersbach et al., 2020) reanalysis, is used to initialize land surface variables for GloSea5 and GloSea6, respectively. GloSea5 and GloSea6 have been used to carry out 60-day (depending on ensemble or variable, 6-month forecast is conducted for the seasonal prediction) retrospective forecasts starting on the 1st, 9th, 17th, and 25th of every month for 26 years (1991–2016) and 24 years (1993–2016), respectively, but evaluations are conducted with 24-year forecasts for the fair comparison between both systems. To operate ensemble forecasts, the Stochastic Kinetic Energy Backscatter (SKEB2; Tennant et al., 2011) and the Stochastic Perturbation of Tendencies (SPT; Sanchez et al., 2016) scheme is used to perturb initial states in GloSea5 and GloSea6, respectively. Compared to the SKEB2, the SPT scheme imposes additional constraints on energy and water conservation, leading to an increase in the ensemble spread without degrading ensemble mean fields, which is especially beneficial over the tropics. Based on these methods, GloSea5 and GloSea6 operate 3 and 7 ensemble forecasts and have been implemented by the KMA in international S2S prediction project for 2020–2022 and 2023–present, respectively. The description of their model configuration is summarized in Table 1.

| | | GloSea5 | GloSea6 |
|---|---|---|---|
| Hindcast period | | 26 years (1991–2016) | 24 years (1993–2016) |
| Ensemble | Method | Stochastic Kinetic Energy Backscatter (SKEB2) | Stochastic Perturbation of Tendencies (SPT) |
| | numbers | 3 | 7 |
| Resolution | Atmosphere | Horizontal: N216 (0.83°×0.56°) Vertical: L85 (~85 km) | |
| Initial conditions | Atmosphere | ECMWF ERA-interim | |
| | Land | JULES offline run (JRA55 atmospheric forcing) | JULES offline run (ERA5 atmospheric forcing) |
| | Ocean/Sea-ice | NEMOVAR (UKMO) | |
| Model physics | Atmosphere | GA6.0 | GA7.2 |
| | Land | GL6.0 | GL8.0 |
| | Ocean | GO5.0 | GO6.0 |
| | Sea-ice | GSI6.0 | GSI8.1 |
| | Coupler | OASIS3 | OASIS3-MCT |

**Table 1: Description of the GloSea5 and GloSea6 model configurations.**

## 2.2 Validation Data

The daily maximum and minimum temperature over land at a height of 2 meters are sourced from NCEP CPC analysis produced by NOAA Physical Sciences Laboratory (PSL; https://psl.noaa.gov). The temperature data have a 0.5° horizontal resolution and are available for 1979–present. The daily mean temperature is acquired by arithmetically averaging maximum and minimum temperature. Hereafter, daily mean, maximum, and minimum temperature will be referred to as Tmean, Tmax, Tmin, respectively.

The ERA5-Land is an offline land reanalysis (Muñoz-Sabater et al., 2021) of the Tiled ECMWF Scheme for Surface Exchanges over Land incorporating land surface hydrology (H-TESSEL) land surface model with four soil layers (0–7-, 7–28-, 28–100-, and 100–289-cm depth), forced by the ERA5 atmospheric reanalysis. ERA5-Land has a horizontal resolution of ~0.18 and an hourly temporal resolution. To enhance the spatial resolution of the ERA5-Land, ERA5 near surface atmospheric variables (e.g., temperature, humidity, and pressure) used for boundary conditions are corrected to account for the altitude difference that came from the lower resolution of ERA5. This study uses ERA5-land as a reference for snow cover extent to diagnose the modelled snow. Compared to the satellite-based datasets, the snow cover is accurately described in ERA5-Land whereas ERA5 is notably overestimated (Kouki et al., 2023). ERA5 assimilates snow depth and cover information from several SYNOP (surface synoptic observation) stations and IMS (Interactive Multisensor Snow and Ice Mapping System) data over the Northern Hemisphere.

*In situ* observations of surface SM are employed to evaluate the model climatological bias and SM memory (SMM) across the globe. International Soil Moisture Network (ISMN; Dorigo et al., 2021) is used to obtain daily mean SM sensed from 5-cm to 10-cm. While flagged measurements classified as "good" quality are used, additional quality control procedures are applied to avoid data redundancy and spurious SM characteristics. First, we exclude the Snowpack Telemetry network (SNOTEL) which has large uncertainty in SM estimates because it is designed to measure snow variables. Second, if observations at one site are made at several depths within that range, it will be represented as a value close to 5-cm. Despite the previous steps, if SM is measured at the same location and depth by different sensors, only one of them is selected to avoid the loss of SM characteristics from simple averaging of many sensors. Lastly, the z-score of SM measured from each sensor is calculated and the sensor with the lowest value is selected. The SM z-score is defined as:

$$Z = \frac{\sum_{t=1}^{N} \frac{X_t - \bar{X}}{\sigma_X}}{\sqrt{\frac{N}{1 + tau}}} \tag{1}$$

where $X_t$, $\bar{X}$, and $\sigma_X$ are the daily time series, timely averaged value, and temporal standard deviation of SM in daily time scale ($t$), respectively. $N$ and $tau$ represent the sample number of daily time series and corrected SMM (described in subsection 3.1), respectively.

A time-filtered satellite product of daily surface SM, originated from the COMBINED European Space Agency (ESA) Climate Change Initiative (CCI) Soil Moisture v06.1 dataset (Dorigo et al., 2017), is used to assess the global SMM simulated by
180 forecast models. Remotely sensed SM datasets inherently contain random and periodic errors, particularly in high-frequency variability, due to the radiometric instrument performance, viewing angle variations, spatial resampling, imperfect parameterizations used in retrieval algorithms, and so on. Due to these errors, the daily time series of satellite-based SM retrieval often shows intervals with an increase in SM without rainfall or any other water supply (see Fig. 6 in Seo and Dirmeyer, 2022a), which is unexplainable by the surface water budget. This erroneous soil moisture behavior hampers the representation
of realistic SM dynamics and land-atmosphere interactions due to a decrease in the SM autocorrelation value. Since the SMM is calculated with the time-lagged SM autocorrelation, assuming that the daily SM time series is exponentially decaying, the inherent error in the satellite data lead to an underestimation of SMM. To avoid the problem, this study uses the time-filtered surface SM product covering 21 years (2000–2020) with 0.25° spatial resolution, using a Fourier transform with LSM datasets (Seo and Dirmeyer, 2022b). The time filtered SM product provides a better representation of the surface SM time series, which
also contributes to the improvement of the SM characteristics (i.e., SM memory and error) compared to the result from *in situ* observations. Hereafter, we refer to the adjusted ESA CCI SM based on the LSM simulations as ESACCI$_{adj.}$

The Global Land Evaporation Amsterdam Model (GLEAM; Martens et al., 2017) provides a dataset of terrestrial heat fluxes and soil wetness derived from algorithms integrating satellite-observed geophysical variables. Based on the Priestley and Taylor equation, GLEAM estimates potential evaporation from net radiation and near-surface air temperature observations.
They are converted into actual evaporation through a multiplicative evaporative stress factor based on observed Vegetation Optical Depth (VOD) and estimated root-zone SM. This study uses the daily surface SM, net radiation, latent heat flux, and sensible heat flux from version 3.5a (https://www.gleam.eu/) covering 21 years (2000–2020) with a 0.25° spatial resolution.

The Multi-Source Weighted-Ensemble Precipitation (MSWEP) version 2.8 is the gauge-, satellite-, and reanalysis-based precipitation dataset used for validation, available from 1979 to the present. The precipitation data have a 0.1° horizontal
resolution and 3-hourly temporal resolution (Beck et al., 2019a). Its superior performance is primarily attributable to the inclusion of daily gauge observations compared with 26 gridded precipitation datasets (Beck et al., 2019b).

**3 Methodology**

This study aims to investigate the impact of an improved snow scheme in the seasonal forecast system on the fidelity of snow
behavior on contemporaneously and during the next warm season after snow melt. Given the many changes between GloSea6 and GloSea5, we cannot attribute all differences in performance to any single change, but we assume changes in the simulation of snow are principally due to the major changes in the snow scheme. To compare model performance between GloSea6 and GloSea5 in the physics of snow freezing and melting, 100-day long retrospective forecasts initiated on the 1$^{st}$ day of October– April spanning 24 years (1993–2016) are used. Although ensemble simulations are carried out in both models, a single member
run is used in this study because 24 yearly samples are sufficient to represent the climatology of the seasonal cycle. The shift

of the snow melting season alters the availability and variability of SM for spring and summer season. 60-day long retrospective forecasts starting on 1st, 9th, 17th, 25th of May–August of 24 years are used to demonstrate the snow effect on the model climatological bias of surface SM, surface air temperature, and precipitation during northern hemisphere warm season when land-atmosphere feedback is most active. Most of the evaluations are accounted for by the fidelity of the modeled land-atmosphere interactions calculated by the daily mean time series of all simulations during boreal summer, thereby representing the climatology of coupling metrics. The ensemble mean values are used in the climatological bias analysis, while the coupling metrics are calculated with each ensemble, and each ensemble result is averaged to avoid the physical correlation between variables fading out in the ensemble-averaged time series. To identify the model improvement with testing statistical significance, 384 initiated forecast runs are validated in each forecast system and tested for statistical significance using a Student's *t* test. Model prediction skill as a function of forecast lead time is not evaluated in this study, because the result is sensitive to the number of ensembles rather than the version upgrade of the forecast model (not shown here).

### 3.1 Soil moisture memory

To evaluate the SM persistence simulated in the model, the autocorrelation-based SMM is employed. First, assuming that the evolution of the daily SM time series follows a first-order Markov process (Vinnikov and Yeserkepova, 1991), the decay frequency ($f$) of SM can be defined by a function of SM autocorrelation ($AR$) at lag day ($\tau$) (Dirmeyer et al., 2016; Seo and Dirmeyer, 2022b). Its formulation is followed as:

$$AR(\tau) = exp(-f\tau) \tag{2}$$

The SMM is defined with an e-folding decay time, at which the autocorrelation of SM drops to $1/e$. By a linear fitting of $ln[AR(\tau)]$, the memory is calculated as the value of $\tau$, when the linear extrapolation between $ln[AR(\tau = 1)]$ and $ln[AR(\tau = 2)]$ is intersected to $ln[AR(\tau)] = -1$. Since the SM behavior is not perfectly fitted on the first-order Markov process, the displacement of the extrapolated linear fit at $\tau = 0$ is defined with the measurement error mostly attributed to random errors (Robock et al., 1995). To measure the SMM under the assumption that there is no measurement error, the extrapolated linear fit is shifted to intersect origin point and the intersected $\tau$ value between the shifted linear fit and $ln[AR(\tau)] = -1$ is the corrected SMM. Time-filtered ESA CCI and modeled SM products exhibit the marginal measurement error (Seo and Dirmeyer, 2022b), so that this study focuses on the improvement in the representation of the corrected SMM in the model simulations. The autocorrelation is calculated by concatenated time series of daily SM over JJA (June–August) of 24 years (1993–2016) with modelled SM time series, but the SMM analysis with the time-filtered satellite dataset is conducted over the 17-year period (2000–2016) due to the data availability. In the calculation of the SMM in both seasonal forecast systems, the SM time series over JJA are concatenated with 30-day forecast time series starting on the 1st of each month, and the time series for each year are further concatenated to produce the 24-year JJA SM time series. The SMM is calculated in each ensemble forecast and represented by the median of the ensemble values.

## 3.2 Time-lagged terrestrial coupling index

To characterize the causality of land-atmosphere interactions, this study adopts time-lagged terrestrial coupling index (LTCI). The original terrestrial coupling index quantitively measures the sensitivity of target variable ($TV$: responding to a feedback) to source variable ($SV$: triggering to a feedback) to demonstrate their physical process connection across a range of time scales (e.g., hourly, daily, monthly, or yearly time series) (Dirmeyer, 2011; Seo and Dirmeyer, 2022a). Based on this matric, the causality of the land-atmosphere feedback is applied by setting a 1-day time lag in the time series of $TV$ compared with $SV$. This is formulated as:

$$LTCI_d(SV_t, TV_{t+1}) = R(SV_t, TV_{t+1}) \times \sigma_{TV_{t+1}} \tag{3}$$

the subscript $d$ refers to using daily time series and $t+1$ denotes 1-day time lag against the raw time series ($t$). $R$ and $\sigma$ represent the temporal correlation coefficient, and the temporal standard deviation, respectively. To explore the quantitative response of precipitation variability to the land surface flux partitioning, this study sets the source and target variables as precipitation ($PR$) and evaporative fraction ($EF = LE/(H + LE)$), respectively, referred to as $LTCI_d(EF_t, PR_{t+1})$.

## 3.3 Methodology to define land coupling regime

This study evaluates model performance in the simulation of land coupling regimes in GloSea5 and GloSea6. Land-atmosphere interaction is controlled by land surface energy and water exchanges. Depending on their relative dominance, water- and energy-limited regimes are categorized, where the flux partitioning between sensible and latent heat flux are controlled by the availability and variability of SM or by net radiation mainly dictated by the atmosphere, respectively. They are separated by a critical value of SM at each location; the dry and wet side of the critical value exhibits water- and energy-limited coupling processes, respectively. Corresponding to the dominant response of the partitioning of land heat fluxes attributed to either the land state or the atmosphere, the direction of land-atmosphere coupling is land-to-atmosphere or atmosphere-to-land, respectively (see Fig. 2 in Seo et al., 2024).

To quantify the strength of land-atmosphere coupling based on either the water- or energy-budget predominance, this study compares the temporal correlation of latent heat flux (the key variable linking water and energy budgets) with the surface SM $[R(SSM, LH)]$ and net radiation $[R(R_n, LH)]$, respectively. Thus, both independent proxies, measuring two distinct land coupling processes, serve as the x- and y-axes in a colour square, and the comparison between them indicates the relative dominance in the definition of land coupling regime (Seo et al., 2024).

## 4 Results

### 4.1 Seasonality of land surface variables

To assess the model performance in simulating snow freezing and melting processes, this study compares the representation of the seasonal cycle of land surface variables between GloSea6 and GloSea5. Although the land initial conditions are generated by different atmospheric forcing in both forecast models, the difference in initiated snow amount appears to be insignificant throughout the entire snow season (Fig. 1a). GloSea5 and GloSea6 simulate the seasonal cycle of snow freezing process over the Eurasian continent similarly regardless of which the snow scheme is used, while snow melts 2 weeks earlier

in the early summer when a single layer snowpack is adopted. For instance, both models consistently simulate a snow peak in March and are initiated with similar snow conditions in that month, but the snow in GloSea5 disappears before June while it persists until early June in GloSea6. The result resembles the snow melting season represented by ERA5-Land. The multi-layer snowpack leads to the lower surface albedo (SF. 3b), which in mid- to high-latitude regions is principally in needleleaf and C3 grass land types defined in the JULES LSM (SF. 2a). In GloSea6, the mapping from the IGBP classification to JULES

land surface types has been refined to improve surface albedo representation. The proportion of bare soil within the grassland, cropland, and crop-natural mosaic the International Geosphere Biosphere Programme (IGBP; Loveland et al., 2000) classes was reduced, as the original mappings likely incorporated elevated bare soil values to represent seasonally barren vegetation (Walters et al., 2019). This adjustment extends the coverage of vegetated land types, notably for C3 grass cover (Wiltshire et al., 2020). Consequently, the shift from bare soil to vegetated surfaces decreases surface albedo, as the expanded vegetation

area penetrates snow cover during the winter season.

Although similar SM states are initialized in both forecast models for the entire analysis period, GloSea5 shows a model forecast drift in the wet direction from October to March, indicating the systemic inconsistency between the initial SM state from the LSM offline simulation and the coupled model climatology (Fig. 1b). Because the snowpack serves as a barrier to energy and water exchange between the land and the atmosphere, later snowmelt delays the onset of evaporation, which slows

the physical process of drying out SM. Thus, the implementation of the multi-layer snowpack results in the climatologically wetter SM following the onset of snowmelt (May and June). The SM difference between GloSea6 and GloSea5 deepens toward the middle of the summer season. In contrast, GloSea6 simulates less soil moisture throughout the snow-covered season, although the initial soil moisture condition is similar in both simulations. The warmer soil temperature in GloSea6, induced by the snow insulation effect, increases the fraction of unfrozen soil moisture. Unlike soil ice, liquid water in the soil remains

mobile, contributing to subsurface runoff and potentially evaporation, resulting in drier soil.

The effect of the multi-layer snow scheme on soil and air temperatures depends on the snow accumulation, snow peak, and snow melting seasons. The snowpack plays the role of limiting transfer of heat between air and soil due to the enhanced insulation (SF. 3e). Therefore, the multi-layer snow scheme provides a stronger insulating effect, simulating significantly warmer soil temperature from snow cover onset through March, when air is colder than the land surface (Fig. 2c). For the

surface air temperature, GloSea6 is colder during the snow freezing season due to the energy loss from the air to the ground

(Fig. 2d). In February and March, when the snow begins to melt, GloSea6 simulates higher air temperature because the snowmelt over warmer ground results in reduced cooling from below (Walters et al., 2019). During the early summer season, the surface cooling in GloSea6 is accounted for by the abundance of SM. Increased partitioning of land heat fluxes to latent heat leads to stronger evaporative cooling.

To illustrate the physical sequence between land surface variables by the realization of snow physics, the time series of major water budget variables is compared between both simulations (Fig. 1e). The surface albedo of GloSea6 becomes larger than that of GloSea5 at the end of March, which results in increased soil moisture about 3 days after. The increase in soil moisture resulting from the reduction in latent heat flux, with a subsequent rise in precipitation begins after the soil moisture increase. The lead-lag correlation between soil moisture and precipitation shows statistically significant values at 0 and +1 lead-lag day

and the 1-day lagged value is the highest (Fig. 1f). In other words, the increased soil moisture in mid-latitude regions likely increases precipitation based on the positive evapotranspiration-precipitation feedback.

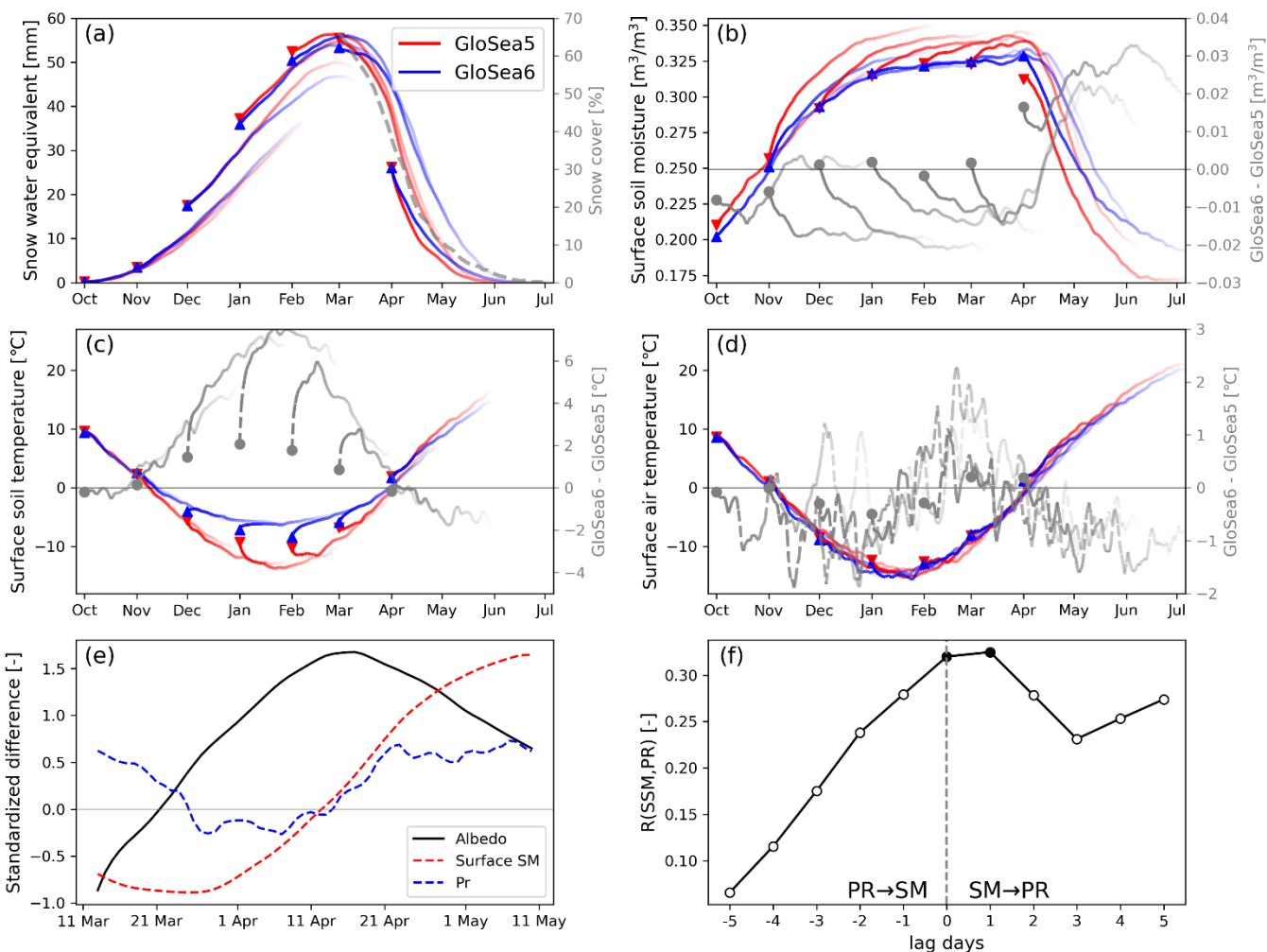

**Figure 1: Climatological seasonal cycle of 24-year (1993–2016) averaged (a) snow water equivalent, (b) surface soil moisture, (c) surface soil temperature, and (d) surface air temperature simulated by GloSea5 (red) and GloSea6 (blue) over the Eurasian continent (0–130E, 45–55N), where 100-day forecast lines fade at increasing lead forecasts and coloured marks indicate initial states on the first day of each month (surface soil temperature shows 60-day forecast due to data availability). To validate the snow melting processes in the model simulations, the grey dashed line in (a) denotes ERA5-Land snow cover. Additionally, to denote the response of surface soil and air temperature to the snow physics scheme, grey dashed lines display the difference between GloSea6 and GloSea5 throughout the snow accumulation and melting seasons. (e) Climatology of 25-day running averaged time series, initiated at each year on 1 March, of the standardized difference (GloSea6-GloSea5) for surface albedo, surface soil moisture, and precipitation. (f) Lead-lag correlation coefficient for the daily time series of the difference between GloSea5 and GloSea6 for surface soil moisture and precipitation with 70-day forecast initiated at each year on 1 March to demonstrate soil moisture-precipitation coupling, where black filled marks denote the correlation value is statistically significant at a 99% confidence level. A positive lagged day indicates that soil moisture leads precipitation, and negative is vice versa.**

## 4.2 Evaluation of model climatological error and bias over the globe

Although soil moisture has historically not been a verifiable quantity in weather forecast models (Koster et al., 2009), the adoption of soil moisture data assimilation makes soil moisture a variable for validation (Seo et al., 2021). To identify the representation of surface SM, this study compares the climatological mean between both forecast models and evaluates their model error against in-situ measurements. The difference in SM simulation between GloSea6 and GloSea5 is large above 40˚N regions across all forecast lead times (Fig. 2a). In particular, the difference is dominant over the snow frontal region, indicating that the difference is related to the additional snow insulating effect in the GloSea6 LSM. Differences at lower latitudes are likely due to other model changes. To assess model fidelity, SM simulated by GloSea5 (Fig. 2b) and GloSea6 (Fig. 2c) are validated against in-situ measurements (mostly distributed over the North America and Europe). Although both models simulate a reliable SM climatology over relatively dry regions (~0.1 $m^3$ $m^{-3}$), modeled SM is systematically underestimated when model values are between 0.1 and 0.2 $m^3$ $m^{-3}$. Most of the underestimated sites are located above 40 N (SF. 4). Although model errors still remain in GloSea6, the drying errors are significantly improved as the SM becomes wetter and the spatial agreement, as measured by the correlation coefficient, is also increased.

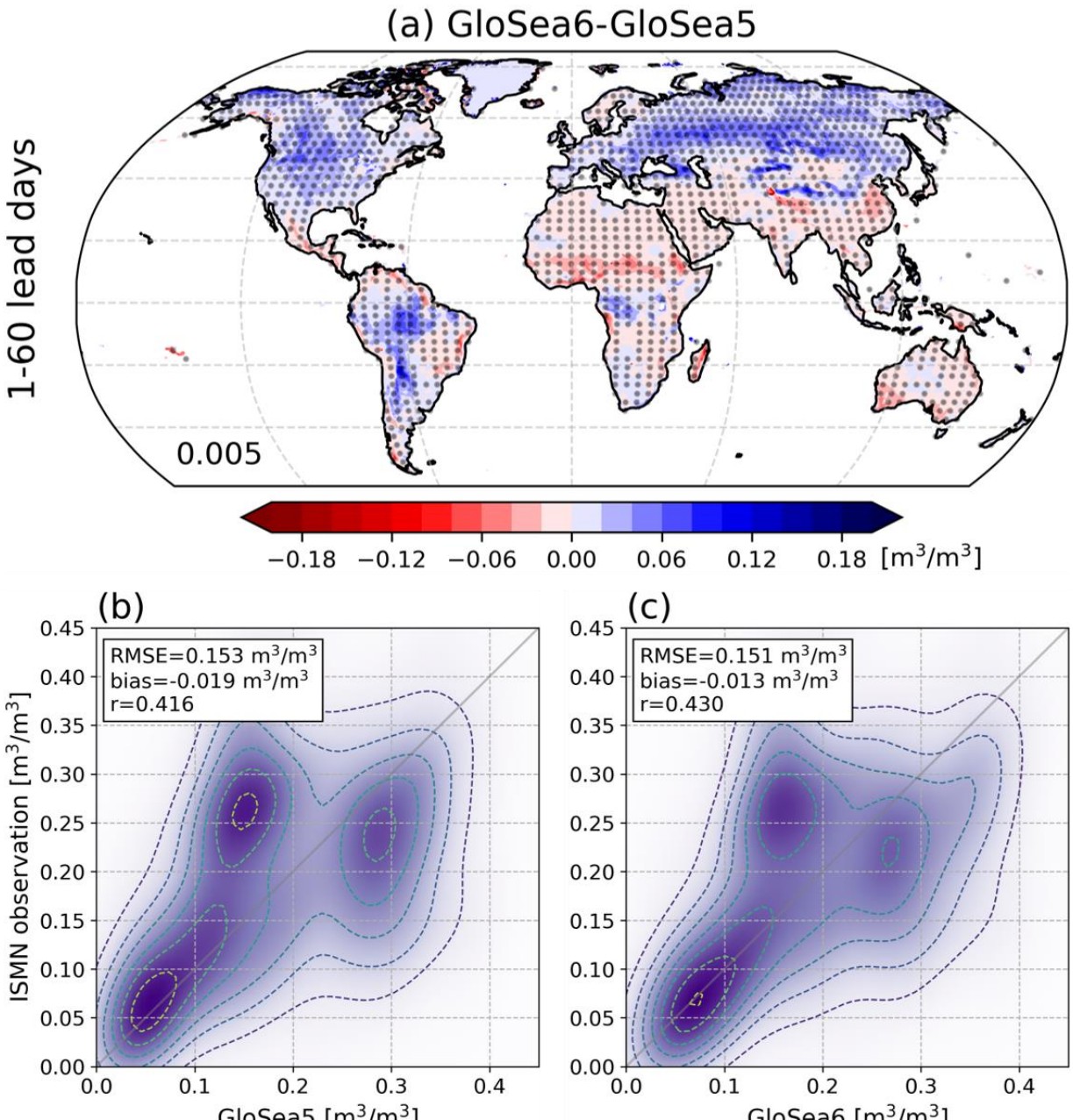

Figure 2: (a) Spatial distribution of climatological surface soil moisture difference between GloSea6 and GloSea5 of the average of 1–60 lead forecast days from the runs initiated in May–August of 1993–2016. The dotted area indicates the difference is statistically significant at a 95% confidence level and global averaged value is indicated in the lower-left corner. 2-dimenssional density of modelled surface soil moisture in (b) GloSea5 and (c) GloSea6 against in situ ISMN observations (1720 measurement sites that are mostly are over North America and Europe as shown in SF. 4), where RMSE, bias, and Pearson correlation coefficient are denoted in the upper-left corner.

Since SMM is a key factor in the subseasonal forecasting because its persistence over a few weeks, model fidelity of SMM is crucial for forecast skill. Because memory is shortened by occurrences of precipitation, it is prolonged where the climate is relatively dry. For instance, SM persistence is relatively short over East Asia where the monsoon flow throughout the summer

season leads to an increasing likelihood of rainfall, accompanying wet soil. The spatial patterns of SMM from ESACCI$_{adj}$, ERA5-Land, and GLEAM are similar (Figs. 3a,b,c), but ESACCI$_{adj}$ is noisy at high-latitudes because SM dynamics are not perceived by the satellite when the surface is frozen. The globally averaged values of SMM from ESACCI$_{adj}$, ERA5-Land, and GLEAM are 10.4, 8.3, 11.5 days, but the Amazon, tropical Africa and Southeast Asia, which have dense vegetation but short SMM, are not sensed by satellite. This likely biases the global SMM estimate from ESACCI$_{adj}$ toward shorter timescales.

The spatial distribution of SMM determined from the observational products is reliably simulated over the globe in GloSea5 and GloSea6. Improvements in SMM bias and spatial agreement are shown in GloSea6 (Figs. 3d,e). The underestimated SMM in GloSea5 is increased by 0.8 days and the spatial correlation of the SMM with the observed fields is also improved. When the assessment is performed with in-situ measurements (SF. 5), the model-based SMM is the better match to the observations (SFs. 5b,c) and there is a significant improvement to simulate the SMM in GloSea6 compared to the GloSea5 (SFs. 5d,e).

When the soil becomes wet due to the late onset of snow melt, the SM decay in response to rainfall is slow, thereby increasing the SMM in mid-latitude regions (Fig. 3e). In contrast, there are some regions (e.g., the southern region of the Amazon, central West Africa, and India) where SMM decreases, the main reason being an increase in rainfall.

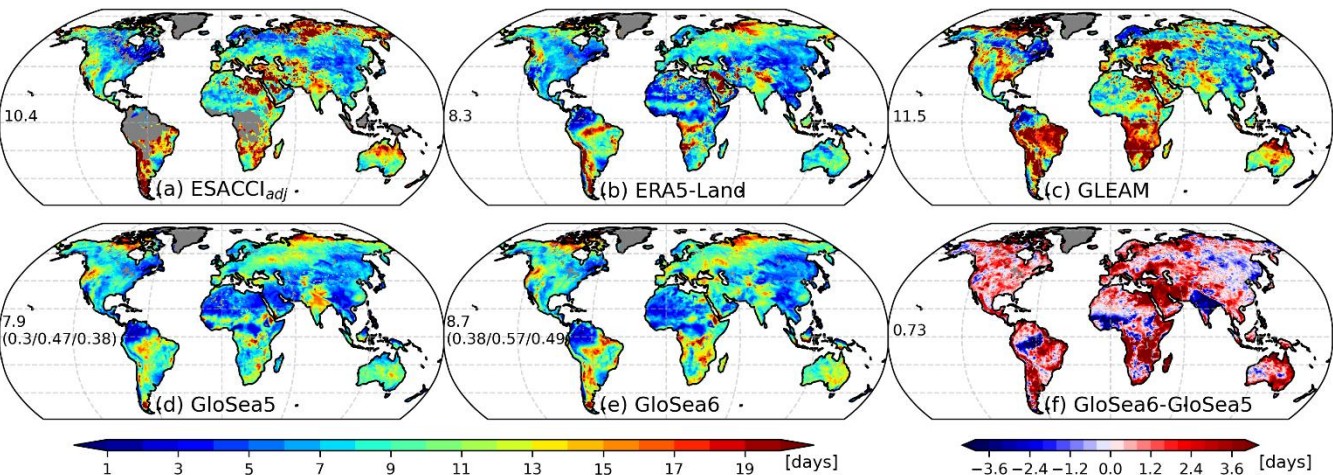

**Figure 3: Surface soil moisture memory from (a) ESACCI$_{adj}$, (b) ERA5-Land, (c) GLEAM, (d) GloSea5, (e) GloSea6, and (f) the**
365 **difference between GloSea6 and GloSea5. Global mean values are denoted in the middle-left in each panel. The bracketed values indicate the spatial correlation of the modelled soil moisture memory compared to ESACCI$_{adj}$ (left), ERA5-Land (middle), and GLEAM (right).**

Features of the simulation of surface air temperature in GloSea6 include reduced bias for daily mean and subdaily time scales across all forecast lead times, explainable by the changes in land physics. GloSea6 represents a decrease in Tmean bias despite

the existence of significant positive bias over North America (Fig. 4b). GloSea6 simulates warmer and colder temperatures over the tropics and mid-latitudes, respectively, compared to GloSea5 (Fig. 4c). To identify the impact of two major

modifications in the LSM on temperature simulation, when the assessment of Tmean is decomposed into the Tmax and Tmin, the results are not consistent with the daily mean. Although the Tmean bias is small in both forecast systems, it results from the cancellation of biases for Tmin and Tmax. Tmax shows a large negative bias north of 50˚N and a positive bias over warm

arid regions (e.g., Southwest Asia) (Figs. 4d,e). Tmin appears to have a large positive bias over the globe, except for the Sahara and Southwest Asia, which have a negative bias (Figs. 4g,h). The effect of the multi-layer snow scheme on forecasting temperature is primarily surface cooling over snow frontal areas throughout the entire day (Fig. 4c), even though the temperature response is more sensitive during the daytime (Figs. 4f,i). This is because there is a larger latent heat flux during the daytime, resulting in a larger evaporative cooling. On the other hand, GloSea6 simulates warmer Tmean over the tropics,

particularly in Tmax, which likely results from updating the land surface albedo as a function of shortwave wavelength.

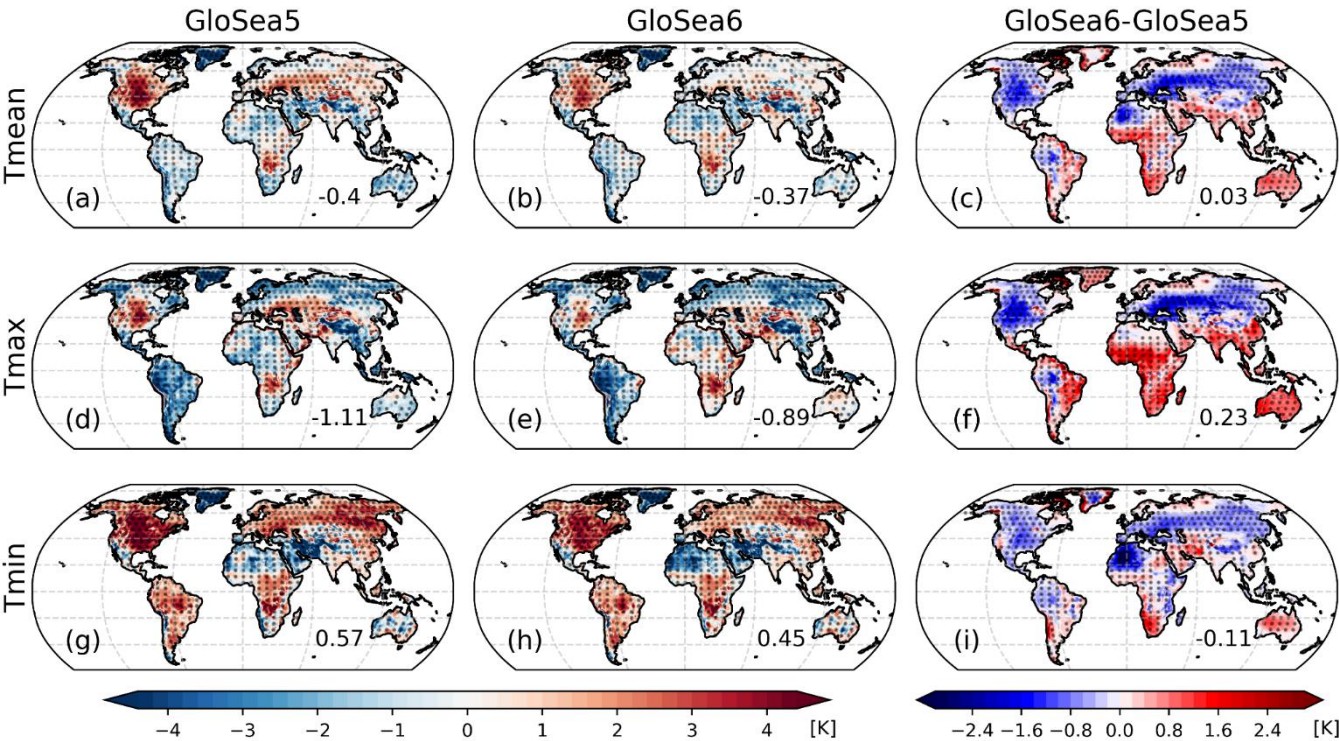

**Figure 4: Spatial distribution of daily mean (upper row; a–c), maximum (middle row; d–f), and minimum (lower row; g–i) surface air temperature bias of the average of 1–60 lead forecast days in GloSea5 (first column), GloSea6 (second column), and the difference between both models (last column). Area averaged bias is denoted in the lower-right corner in each panel. Dotted areas indicate the**
**bias is statistically significant at a 95% confidence level.**

The systemic error of surface air temperatures, measured by root-mean-square error (RMSE), is further investigated using 60-day lead forecasts. In general, the error in Tmean, Tmax, and Tmin from GloSea6 is largely reduced compared to that from GloSea5. In particular, GloSea5 shows a large Tmean RMSE over the eastern US, Siberia, and Australia (Fig. 5a), but the error is significantly mitigated in GloSea6 (Fig. 5c). Tmean errors in the eastern US and Siberia are influenced by both Tmax and

Tmin. Based on the temperature bias analysis, this result is attributed by the improvement in the snow scheme that has effects

throughout the day. Over Australia, the decrease of Tmean error is mostly influenced by Tmax, which is accounted for by the improvement of land-atmosphere interactions during the daytime (cf., Fig. 10d). However, some errors are aggravated in GloSea6. For instance, in northeastern Eurasia, Tmax RMSE is significantly increased by an exacerbated cold bias, which is related to a cold bias in initial conditions (not shown). The multi-layer snowpack reinforces this bias in GloSea6.

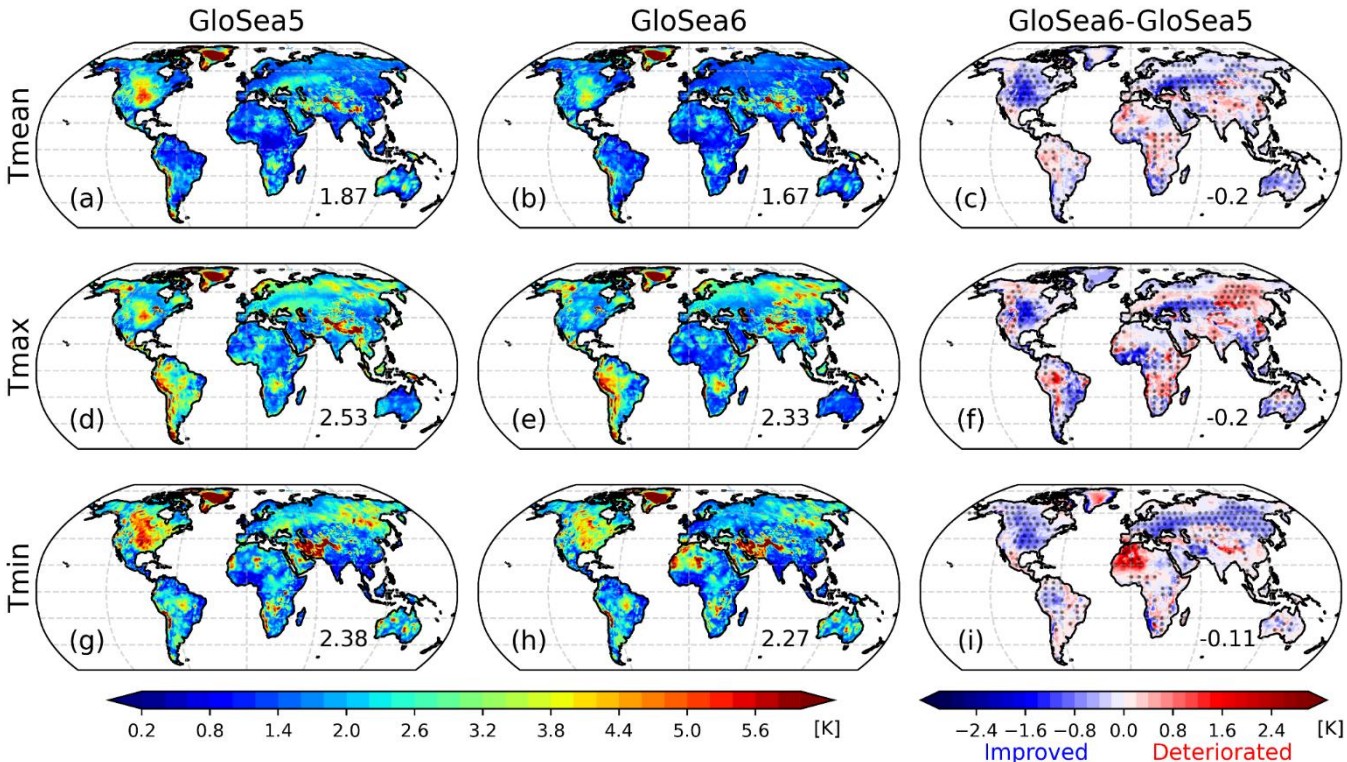

**Figure 5: Same as Fig. 4, but for RMSE of surface air temperature variables. Blue and red shading in difference maps (c, f, and i) indicate the improved and deteriorated forecast performance of GloSea6, compared with the GloSea5.**

Model performance in simulating precipitation is also evaluated in GloSea5 and GloSea6. Both models show an overestimation of precipitation across the globe because of the wet bias over South America, central Africa, southern China, and northeastern

Eurasia (Figs. 6a,b). Although the globally averaged bias increases in GloSea6, this is largely due to a reduction in the negative bias over the continental United States (CONUS) and western and central Eurasia, as the positive bias is amplified or maintained in areas that have wet biases in GloSea5 (Fig. 6c). The increased precipitation over the mid-latitude regions is explained by the abundant SM from snow melting process under positive evapotranspiration-precipitation feedbacks (cf., Fig. 7). The precipitation errors of GloSea5 and GloSea6 appear to be spatially large over the areas where the mean precipitation

climatology is high (e.g., East America, Central America, South and East Asia, and Central Africa) (Figs. 6d,e). The difference of precipitation RMSE maps between GloSea6 and GloSea5 reveals a significant improvement in the simulation of precipitation over central CONUS, western and central Eurasia, Central Africa, and South Asia (Fig. 6f). Although entire

regions where the error is reduced cannot be explained solely by advances in land processes, the improvement in the mid- and high-latitude regions of the Northern Hemisphere is likely due to the improved snow physics.

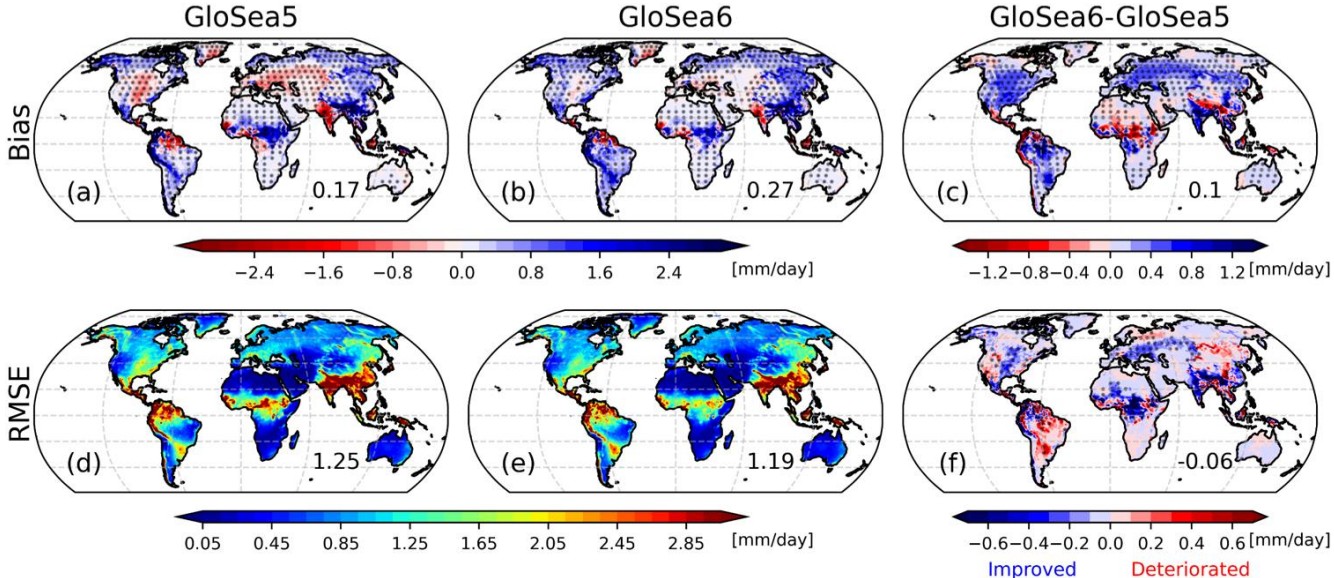

**Figure 6: Spatial distribution of daily mean precipitation bias (upper row; a–c) and RMSE (lower row; d–f) of 60 days forecast in GloSea5 (first column), GloSea6 (second column), and the difference between both models (last column). Dotted areas indicate that bias (a, b, and c) and RMSE (f) are statistically significant at a 95% confidence level.**

To demonstrate the impact of land-atmosphere interactions on the model ability to simulate precipitation, this study assesses the time-lagged terrestrial coupling index ($LTCI$). The observed $LTCI_d(EF_t, PR_{t+1})$ generally represents a positive coupling to precipitation over the globe due to the positive correlation $R(EF_t, PR_{t+1})$, with particularly strong feedbacks over the areas where precipitation variability $\sigma_{PR_{t+1}}$ is high (e.g., Central America, Eastern CONUS, South Asia, and East Asia) (Fig. 7a). The spatial pattern of $LTCI_d(EF_t, PR_{t+1})$ simulated by GloSea5 and GloSea6 is similar to the observed distribution, whereas there is an overall overestimation of coupling strength (Figs. 7b,c). Both models commonly overestimate the $LTCI_d(EF_t, PR_{t+1})$ over the Americas (except for the Amazon), northern Eurasia, and South Asia, but the positive bias is mitigated in GloSea6 (Figs. 7e,f). For GloSea6, the large positive bias over the tropics, also simulated by GloSea5, is significantly reduced, attributable to the decrease in $R(EF_t, PR_{t+1})$, while the positive bias over high-latitude regions is slightly amplified by increased $\sigma_{PR_{t+1}}$ (Fig. 7d). The increased variability of daily precipitation in GloSea6 is associated with an increase in mean precipitation (cf., Fig. 6c).

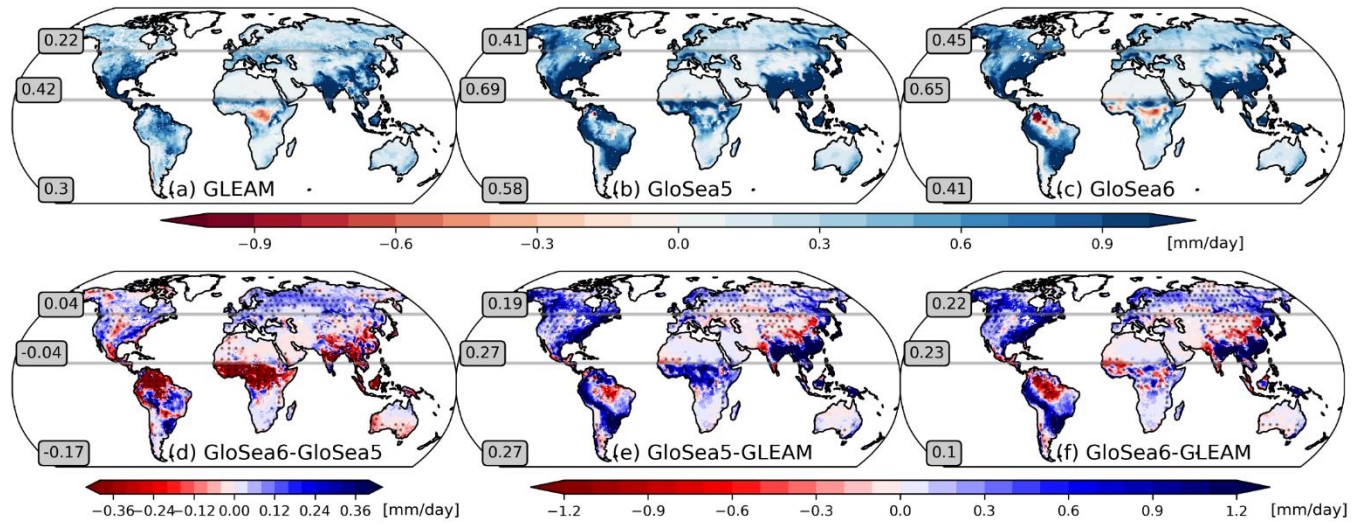

**Figure 7:** Spatial distribution of $LTCI_d(EF_t, PR_{t+1})$ of 60-day forecasts in (a) GLEAM, (b) GloSea5, and (c) GloSea6. (d) The difference between GloSea6 and GloSea5 and the bias of the 1-day lagged TCI simulated by (e) GloSea5 and (f) GloSea6 are displayed, where the dotted areas indicate statistical significance at a 95% confidence level. In each panel, grey horizontal lines isolate three areas (bottom: 60S–15N, middle: 15–50N, and top: 50–90N) and area averaged values is denoted within grey shaded box.

## 4.3 Representation of land coupling processes

The exchanges at the land surface are constrained by the water and energy balance equations, and the strength of water- versus energy-limited processes is quantified by the temporal correlation coefficient of latent heat flux to surface SM or net radiation, respectively, as described in subsection 3.3. In Figure 8, the colour square consists of $R(SSM, LH)$ and $R(R_n, LH)$ on the x- and y-axis, respectively, indicating the relative dominance of water- and energy-limited coupling. The spatial pattern of the GLEAM land coupling regimes is similar to the distribution of the SM climatology, such that water-limited processes are pronounced over climatologically dry areas and vice versa. The classification of the land coupling regime results from the synthetization of the spatial pattern of $R(SSM, LH)$ (Fig. 9a) and $R(R_n, LH)$ (Fig. 10a). The kernel density plot of $R(SSM, LH)$ is bimodal, with clearly separated peaks on either side of zero, while there is a double peak in $R(R_n, LH)$ with a broad peak centered near zero and a pronounced positive peak. For instance, the spatial distribution of $R(SSM, LH)$ and $R(R_n, LH)$ is a zonal dipole structure over CONUS but is meridionally banded over Eurasia.

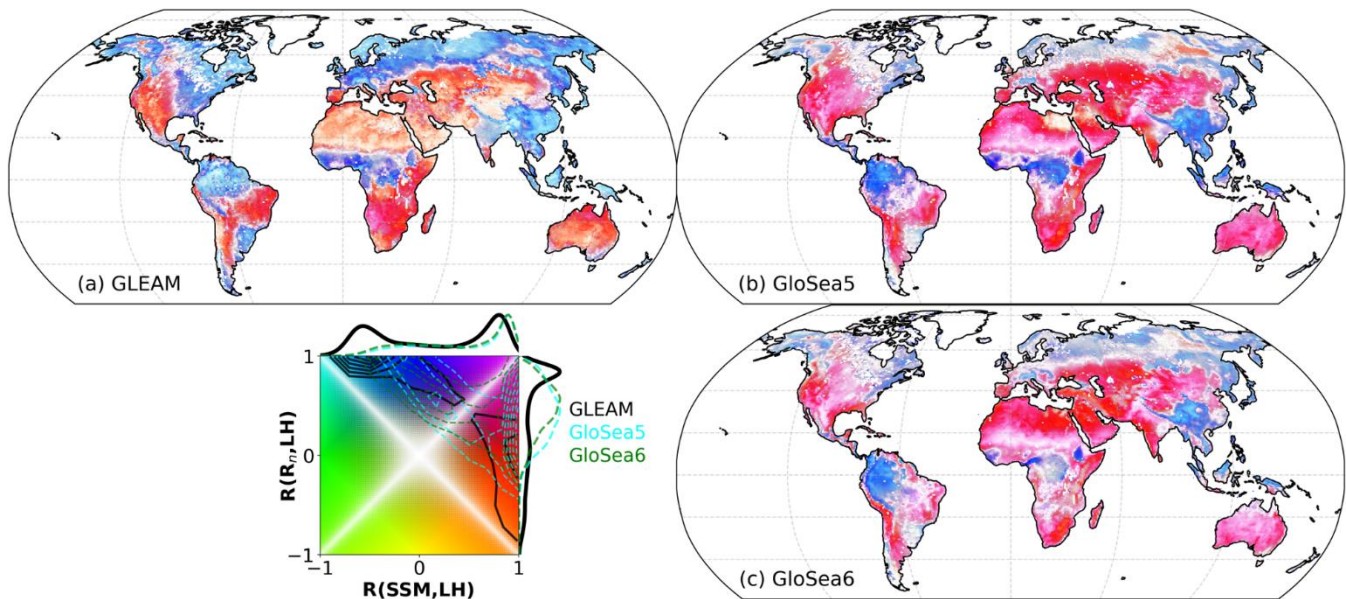

**Figure 8: Spatial distribution of land coupling regime in (a) GLEAM, (b) GloSea5, and (c) GloSea6. Shadings indicate correlations indicated in the coloured square: latent heat flux to surface soil moisture (x-axis) and net radiation (y-axis). The global frequency distributions from GLEAM (black), GloSea5 (cyan), and GloSea6 (green) are shown in the lower-left 2-dimensional coloured square. Their kernel density estimations are along the edges of the coloured square, where each curve has been normalized for the same maximum value.**

GloSea5 and GloSea6 show a single peak on the positive side of the kernel density estimation of $R(SSM, LH)$, which is explained by an overall overestimation of $R(SSM, LH)$ resulting in the expansion of water-limited areas and the degradation of the spatial characteristics in the observation (Figs. 9b,c). The strength of the water-limited coupling is overestimated over the globe, but the positive bias is particularly evident over high-latitude regions (Figs. 9e,f). Nevertheless, the difference between the two forecast system kernel density estimates of $R(SSM, LH)$ is not significant. This does not indicate that the spatial distributions are the same, but rather a cancellation of the changed areas of increase and decrease of $R(SSM, LH)$ in GloSea6 over the tropics and high-latitude areas (Fig. 9d), respectively.

On the other hand, both forecast models show a single peak on the positive side of the kernel density estimation of $R(R_n, LH)$, even though the underestimated energy-limited coupling strength in GloSea5 is greater in GloSea6. The spatial distributions of $R(R_n, LH)$ simulated by the two models similarly underestimate the spatial dependency (Figs. 10b,c), compared to the GLEAM. For instance, in GLEAM, dry and high-latitude regions show negative and large positive values of $R(R_n, LH)$, respectively, but the models reveal positive and negative biases for each region (Figs. 10e,f). Nevertheless, GloSea6 significantly increases the energy-limited coupling strength, which mitigates the negative bias of $R(R_n, LH)$, especially over the high-latitude areas, whereas the underestimation still exists (Fig. 10d). Because the late onset of snowmelt leads to wetter SM over the mid- to high-latitudes during the warm season, the wetter SM climatology limits the sensitivity of land heat fluxes to SM variability, leading to a regime shift of land coupling from water-limited to energy-limited processes.

As a result, GloSea5 and GloSea6 have a limited ability to simulate the observed land coupling regime distributions; the comparison of the 2-dimentional density function for GloSea5 (cyan line) and GloSea6 (green line) in the coloured square. While the water-limited coupling is generally overestimated in both forecast models, the improvement of the energy-limited process in GloSea6 leads to a better classification of the land coupling regime over the globe (Fig. 8b,c). For instance, GloSea5 has an excessive area of red-coloured grid points, indicating the relative dominance of water-limited coupling, while GloSea6 better simulates the spatial pattern of land coupling regimes. In particular, the zonally and meridionally classified dipole pattern over the CONUS and the snow frontal area of Eurasia, respectively, become clear.

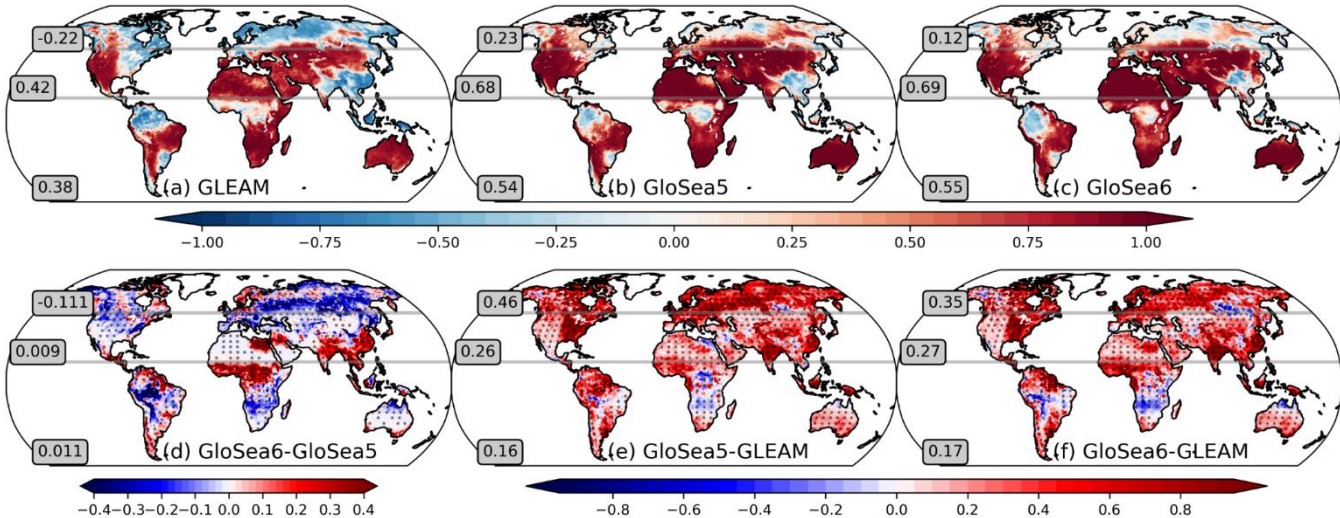

Figure 9: Same as Fig. 7, but for the correlation coefficient between daily latent heat flux and surface soil moisture, to illustrate water-limited processes.

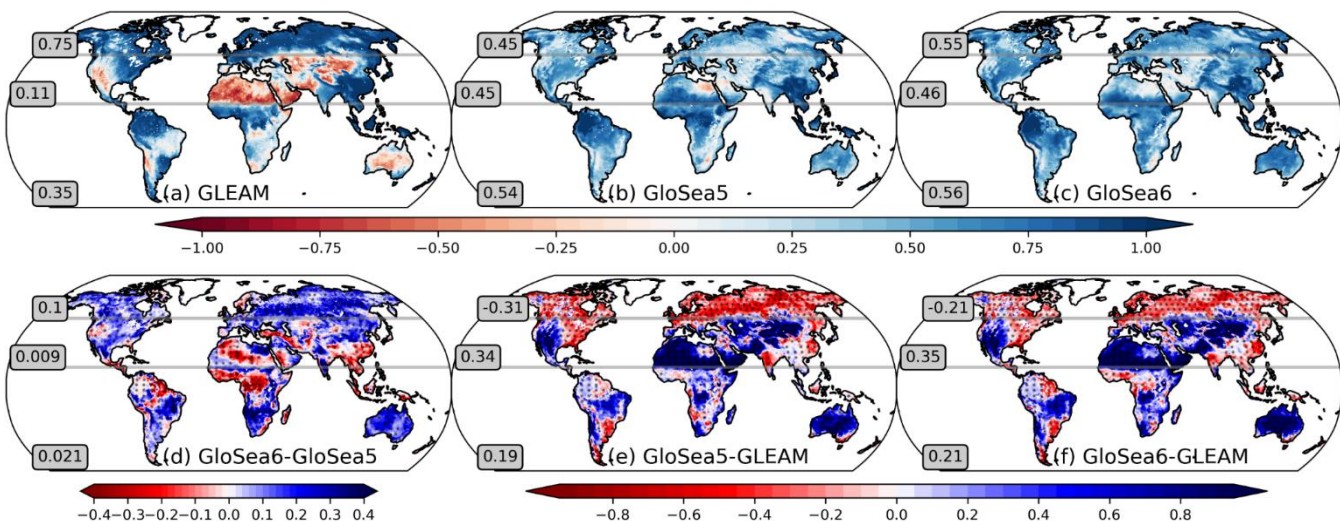

Figure 10: Same as Fig. 9, but for the correlation coefficient between daily latent heat flux and net radiation, to illustrate energy-limited processes.

## 5 Summary and Conclusions

Some land surface models have employed a single layer snow scheme that insulates the near-surface atmosphere from direct access to the heat in the ground. While effective for very thin snow cover, such a scheme fails to simulate the true insulating
effect of the snowpack by prohibiting energy transport between land and atmosphere in deeper snow. This study primarily explores the impact of implementing a multi-layer snow scheme on the climatological bias of the seasonal forecast system. Two sets of the GloSea global retrospective seasonal forecasts over 24 years (1993–2016), from the latest version (GloSea6) and its predecessor (GloSea5), which implement the multi-layer and single layer snow schemes, respectively, are examined to elucidate the role of the insulating effect of snow. The improvement in the model simulations appearing in areas with high
snow variability can be understood as the effect of the multi-layer snow scheme. However, the differences between GloSea5 and GloSea6 in areas unrelated to snow (e.g., South and East Asia, Central Africa, South America, and Australia) are likely the result of various other factors arising from other modifications as part of the model version update. For instance, over India, the bias of too much precipitation over the ocean and too little precipitation over the subcontinent has been improved by updates including the stochastic physics, convection scheme, and warm rain microphysics in the GloSea6 (Walters et al., 2019).
Furthermore, the inclusion of the stochastic physics contributes to a reduction of the precipitation bias over Africa.

The improved snow physics with a multi-layer snowpack better captures the observed snow dissipation season (Fig. 1a) and affects land and near-surface variables throughout the snow accumulation and melting seasons. The land surface warming and cooling due to the insulating effect of the snowpack during the snow peak and melting seasons (Fig. 1c) results in a late onset of snow melt and wetter SM during the following summer season, especially in mid- to high-latitude regions (Fig. 1b and 2a),
leading to reduced error in surface SM (Figs. 2b,c). The changes in land surface processes also affects land surface characteristics, e.g. SM memory is generally increased, which reduces model error in the memory and improves spatial agreement compared to the observational analysis (Fig. 3). Moreover, the greater SM from the advanced snow physics leads to a decrease in temperature with evaporative cooling throughout the entire day (Fig. 4) and an increase in the likelihood of precipitation explained by evapotranspiration-precipitation feedbacks (Fig. 6). The climatological mean shift in temperature
and precipitation through implementing the multi-layer snow scheme in GloSea6 significantly reduces the error in the mid- and high-latitude regions, as the reduced temperature and increased precipitation offset GloSea5's climatological warm and dry bias. On the other hand, the other physics update in the LSM is the land surface albedo, which now varies with shortwave wavelength. Its effect is significant only during the daytime over the tropics, because the effect of surface albedo on the surface energy budget is dominant during daytime.
The spatial distribution of the land coupling regime is similar to that of the SM climatology, with the majority of water- and energy-limited coupling occurring over relatively dry and wet soils (Fig. 8). Assessment of model performance is critical to understanding the issues associated with land-atmosphere coupling processes. Comparing the land coupling regime simulated by GloSea5 and GloSea6, the increased SM in GloSea6 alters land-atmosphere interactions, limiting the strength of water-

limited coupling (Fig. 9) along with enhance energy-limited processes (Fig. 10). Although the relative dominance of water-
limited coupling is still overestimated in both GloSea5 and GloSea6, this problem is corrected over mid- and high-latitude
regions when the multi-layer snow scheme is implemented. The increased SM due to the late onset of snowmelt restricts water-
limited coupling. This results in an increase in $R(R_n, LH)$ complemented by a decrease in $R(SSM, LH)$.

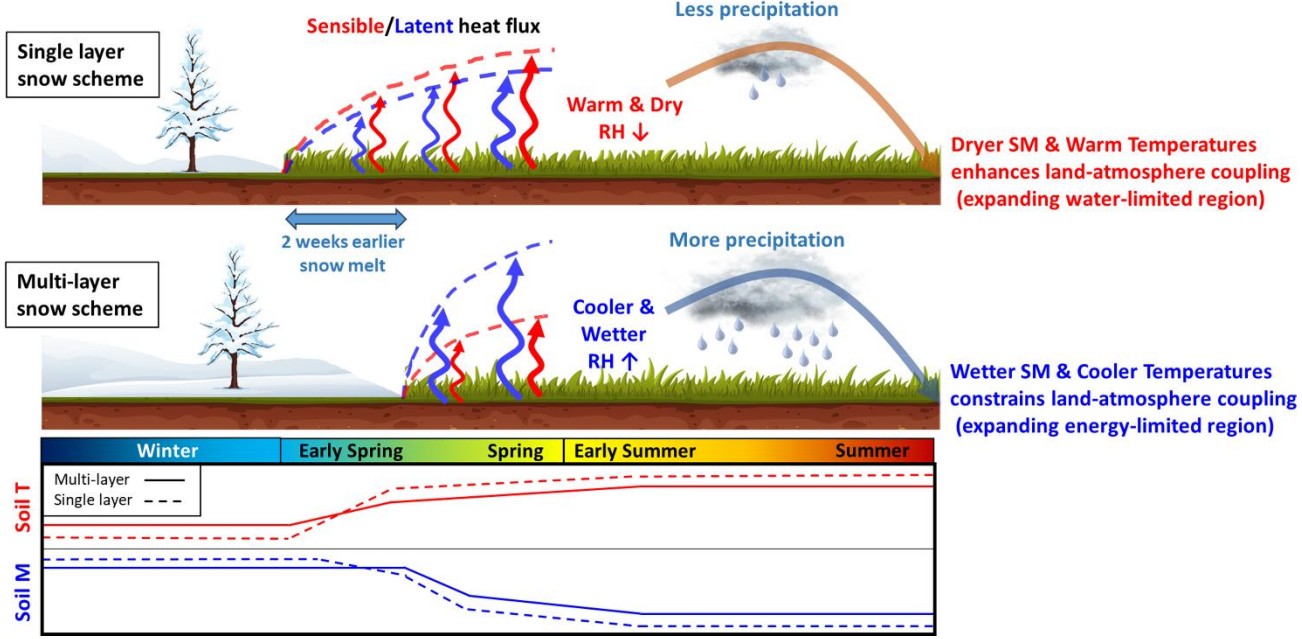

**Figure 11: Schematic of the impact of multi-layer snow scheme on seasonal forecast system from winter through the following**
**summer.**

Because the simulation of realistic snow states affects the water and energy budgets not only in winter also in spring and
summer (Fig. 11), the realization of snow characteristics should be a priority in the process of developing a model. For instance,
if the land surface model is modified to improve land processes for the warm season, when land-atmosphere feedbacks are
evident, without any assessment and improvement of snow behaviour, the model is likely to have a larger error, even if the
snow is simulated realistically. Note that the climatological improvements do not imply an increase in the predictability of
forecast systems, as the increase in forecast skill of temperature and precipitation in GloSea6 is primarily due to the larger
ensemble size (SFs. 6 and 7). Therefore, this study suggests that the implementation of a multi-layer snow scheme is necessary
to simulate the realistic land surface processes in dynamical forecast systems on the subseasonal to seasonal time scale. From
a climate perspective, as global warming increases the variability and uncertainty of modelled snow, reliable future projections
for climate change can be presented with the results of selective use of models that are able to simulate realistic snow
characteristics.

**Acknowledgements**

This study was supported by Korea Meteorological Administration Research and Development program under grant RS-2023-00241809. Eunkyo Seo was supported by Learning & Academic research institution for Master's·PhD students, and Postdocs (LAMP) Program of the National Research Foundation of Korea (NRF) grant funded by the Ministry of Education (RS-2023-000301702). We also wish to thank Sunlae Tak for sharing retrospective forecast datasets which helped us perform the model evaluation.

**Code availability**

The MetUM is available for use under licence. The source code for the Met Office Unified Model (MetUM) cannot be provided due to intellectual property right restrictions. For further information on how to apply for a licence, see https://www.metoffice.gov.uk/research/approach/collaboration/unified-model/partnership. The source code used in the model evaluation of this study is shared on the GitHub (https://github.com/ekseo/Multi-layer_snowpack_GloSea.git, last access: 31 May 2024; https://doi.org/10.5281/zenodo.11243938, Seo, 2024).

**Data availability**

The Copernicus Climate Change Service (C3S) provides access to ERA5-Land data freely through its online portal at https://cds.climate.copernicus.eu/cdsapp#!/dataset/10.24381/cds.e2161bac?tab=overview. ISMN soil moisture observation is publicly available through its online website at https://ismn.earth/en/. CPC Global Unified Temperature data is provided by the NOAA/OAR/ESRL PSL, Boulder, Colorado, USA, can be downloaded from their website at https://psl.noaa.gov/data/gridded/. MSWEP precipitation dataset can be accessed at https://www.gloh2o.org/mswep/. GLEAM data is publicly available at the website: https://www.gleam.eu/. The ECMWF provides access to GloSea6-GC3.2 hindcast data freely through its online portal at https://apps.ecmwf.int/datasets/data/s2s/. Other GloSea retrospective datasets and time-filtered ESA CCI SM product used in this study are available upon request from the authors.

**Author contributions**

ES led manuscript writing and performed most of the data analysis. PD contributed to the interpretation of results and manuscript writing.

**Competing interests**

The authors have no competing interests to declare.

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
