# Peer review of "Improving land-atmosphere coupling in a seasonal forecast system by implementing a multi-layer snow scheme"

_EGUsphere, 2024_

## Author Comment (AC1)

**Reviewer #1**: The paper by Seo and Dirmeyer, titled "Implementation of Multi-layer Snow Scheme in Seasonal Forecast System and Its Impact on Model Climatological Bias," investigates the effects of implementing a multi-layer snow scheme on the climatological biases of a seasonal forecast system. Traditional single-layer snow schemes in land surface models often inadequately capture the insulating effects of snowpack, leading to warm and cold biases during winter and snow melting seasons. The study compares the performance of the Global Seasonal Forecast System (GloSea) versions 5 (single-layer) and 6 (multi-layer) over a 24-year period. Findings reveal that the multi-layer snow scheme in GloSea6 shifts the snow melting season by two weeks, improving surface temperature, permafrost extent, and overall model climatology. This enhancement mitigates near-surface warming bias and improves precipitation simulation over snow-covered regions.

➔ *We thank the reviewer for the comments. We hope we have adequately clarified our descriptions and addressed the points raised.*

However, it overlooks critical differences in vegetation treatment between the Noah and Noah-MP models. Suzuki and Zupanski (2018, doi: https://doi.org/10.1007/s11707-018-0691-2) provide a thorough examination of the uncertainties in solid precipitation and snow depth prediction, which is highly relevant to this study. The differences between the land surface models are notable: the Noah model uses a one-canopy layer with a simple canopy resistance and a linearized energy balance equation representing the combined ground-vegetation surface, considering seasonal LAI and green vegetation fraction. In contrast, the Noah-MP model includes snow interception features such as loading-unloading, melt-refreeze capabilities, and sublimation of canopy-intercepted snow, along with a detailed representation of radiation transmission and attenuation through the canopy, within- and below-canopy turbulence, and different options for representing the biophysical controls on transpiration. Therefore, the changes affect not only snow-covered areas but also the global vegetation albedo and surface temperature. In their results, they report that the snow depth changes, but the snow water equivalent does not. The reason for the longer period of snow cover is believed to be due to the more accurate representation of radiation and turbulent fluxes beneath the vegetation canopy. Therefore, the multi-layer snow model is not the critical factor in this case.

➔ *We agree that vegetation treatment is also a critical factor influencing snow physics in seasonal forecast systems. Compared with the Noah LSM, the improvement of snow simulation in the Noah-MP is due not only to the implementation of multi-layer snow scheme, but also to a semi-tile subgrid scheme to separate vegetation and bare soil. However, this study has used JULES LSM, which is a community model developed by UK Met Office, rather than Noah and Noah-MP LSMs. We have tried to correct any of confusion that may arise regarding the use of land surface model.*

*In both GloSea5 and GloSea6 models, there are no changes to vegetation treatment. The surface of each land grid box subdivided into five types of vegetation (broadleaf trees, needle-leaved trees, temperate C3 grass, tropical C4 grass and shrubs) and four non-vegetated surface types (urban areas, inland water, bare soil and land ice). Regarding Leaf Area Index (LAI), the ancillary parameters are derived from satellite data processed to be consistent with these land cover and plant functional type classifications.*

To enhance the completeness of your study, it is crucial to discuss the impact of vegetation treatment in addition to the multi-layer snow scheme. By addressing these points, the manuscript will provide a more holistic view of the improvements in seasonal forecast systems and their broader climate implications.

➔ *In this study, there is no modification of vegetation in GloSea5 and GloSea6 runs, but the model configuration for the vegetation is now briefly described in the manuscript. Thus, the information of prescribing vegetation according to plant function types has been added in Lines 123-127*

**"Other land surface physics are consistent in GloSea5 and GloSea6. For land surface types, five vegetation (broadleaf trees, needleleaf trees, C3 grasses, C4 grasses and shrubs) and four non-vegetated surfaces (urban, open water, bare soil and permanent land ice) are classified and the monthly climatology of leaf area index, derived from MODIS satellite product (Yang et al., 2006), is prescribed corresponding to the plant functional types. Snow is present on every land tile, including inland water when its temperature is below freezing."**

\* *Yang, W., Tan, B., Huang, D., Rautiainen, M., Shabanov, N. V., Wang, Y., Privette, J. L., Huemmrich, K. F., Fensholt, R., and Sandholt, I.: MODIS leaf area index products: From validation to algorithm improvement, IEEE Transactions on Geoscience and Remote Sensing, 44, 1885-1898, 2006.*

Specific Comments

Introduction and Background: Please include a discussion on the handling of vegetation in land surface models, specifically contrasting the Noah and Noah-MP models.

Methodology: Please provide detailed descriptions of the Noah and Noah-MP models, focusing on their treatment of vegetation and snow processes. In addition, please discuss how these differences might affect your results and the broader implications for climate modeling.

Results and Discussion: Please analyze the impact of vegetation treatment on your findings, especially in terms of global vegetation albedo and surface temperature.

Conclusion: Please emphasize the importance of considering both snow and vegetation processes in land surface models.

➔ *Both forecast models (GloSea5 and GloSea6) have used the JULES LSM, but several updates in the model physics are implemented. Neither Noah nor Noah-MP are part of the model configurations used here. Therefore, we try to reflect only the reviewer's suggestion for describing the vegetation treatment in the JULES LSM.*

---

## Author Comment (AC2)

**Reviewer #2**: The paper "Implementation of Multi-layer Snow Scheme in Seasonal Forecast System and Its Impact on Model Climatological Bias" evaluated the retrospective seasonal forecast performance of the Global Seasonal Forecast System (GloSea) version 5 (GloSea5, with a single-layer snow scheme) and version 6 (GloSea6, with a multi-layer snow scheme) over a 24-year period (1993-2016), focusing on the impacts of multi-layer snow scheme (GloSea6) versus single layer snow scheme (GloSea5) on the climatological biases of the seasonal forecast system. Results revealed that GloSea6 more accurately captures the snow phenology, elongating the snow melting season by two weeks, which improves the simulations of soil moisture, surface temperature, surface evaporation and subsequent land-atmosphere coupling regime in mid-to-high latitudes. This enhancement mitigates near-surface warming bias and improves precipitation simulation over snow-covered regions during late spring to summer. The authors attributed this improvement to the multi-layer snow scheme in GloSea6, yet more analyses are necessary to exclude other model physics updates (including atmosphere, ocean and sea ice) to support this conclusion.

➔ *We thank the reviewer for the comments. We hope we have adequately clarified our descriptions and addressed the points raised.*

Major points:

Snow cover is important in land surface modeling, besides the treatment of snowpack in single layer or multi-layer scheme, snow surface albedo and snow cover fraction (the percentage of a model grid that is covered by snow) are pivotal factors that influence the accumulation and melting of snow cover in climate models. How about the difference in these two factors in GloSea5 and GloSea6?

➔ *As reviewer pointed out, snow cover fraction is also a pivotal factor that balances energy budgets at the land surface by influencing snow surface albedo. However, snow fraction is not included in the list of standard model output, so that it is alternatively estimated by the surface albedo calculated by upward and downward shortwave radiation at the surface. Below, figures (added to the supplementary figure 1) show that GloSea6 simulates more snow in early March, and surface albedo increases around the end of March, compared with the GloSea5. It results from upward shortwave radiation rather than downward component. In other words, increasing snow amount leads to an increase of surface albedo due to higher fractional snow cover at the land surface about 10 days later. The description about surface albedo has been added in Lines 265-267 with updated Fig. 1.*

➔ ***"The multi-layer snowpack also extends the area of snow cover, which leads to the increased surface albedo, where increasing snow amount leads to an increase of surface albedo at the land surface about 10 days later (SFs. 1a,b)."***

[Figure]

Why are surface soil moisture (SSM) simulated in GloSea5 are more than those in GloSea6 from October to March but the situation reversed (i.e., GloSea5 SSM less than GloSea6) after April, although the initial snow amount are close to each other on April 1st (Figure1a,b)?

➔ *Multi-layer snowpack, which has a role of soil insulator, increases the soil temperature during winter season due to inhibiting thermal energy exchange between land and atmosphere, which results in the reduction of energy transport from land to the atmosphere. The warm soil temperature increases the ratio of unfrozen soil – this liquid water is mobile in the soil matrix, unlike soil ice, so that less soil moisture is simulated in GloSea6 from October to March. To demonstrate this process, we carried out two sets of JULES offline simulation using single or multi-layer snow scheme for 24 years (2000-2023), where near-surface atmospheric forcing variables are utilized by ERA5 reanalysis, but the precipitation is used from IMERG data. The bottom figure represents the climatological annual cycle of surface soil temperature and frozen and unfrozen moisture contents over the Eurasian continent (0–130E, 45–55N). The result exhibits the multi-layer snow scheme leads to increased soil temperature and the larger partitioning of unfrozen soil moisture along with more total content of soil moisture. The description is added in Lines 274-277*

*"In contrast, GloSea6 simulates less soil moisture throughout the snow-covered season, although the initial soil moisture condition is similar in both simulations. The warmer soil temperature in GloSea6, induced by the snow insulation effect, increases the fraction of unfrozen soil moisture. Unlike soil ice, liquid water in the soil remains mobile, contributing to subsurface runoff and potentially evaporation, resulting in drier soil."*

[Figure]

In line 270-271, the author claimed that the weaker insulating effect of the single-layer snow scheme leads to warmer surface temperature during thin (snow melting or freezing season) snow cover (figure 1c), in fact, during the freezing season from October to January when the air is colder than land surface, if the single-layer snow scheme provides a weaker insulating effect, it should lead to colder rather than warmer surface temperature. How to understand this contradiction?

➜ *Simply noting surface temperature confuses classifying surface soil and air temperature, so we add both results in Fig. 1c and 1d, respectively. During snow melting season, soil and air temperature in GloSea6 commonly become colder because the later onset of snow melting leads to abundant soil moisture. For the snow season, GloSea6 simulates warmer soil temperature and colder air temperature, which refers to that the multi-layer snow scheme hinders energy transport between near-surface atmosphere and the soil. During the snow peak season, the snow insulating effect also contributes to the increase of surface air temperature, but the direction is opposite. The warmer air during early spring cannot lose its heat to the soil due to insulation by the snowpack, allowing the air to remain warm. The main text has been modified to clarify the soil and air temperature responses in GloSea6 throughout snow freezing, peak, and melting seasons. The description is added in Lines 280-285*

**"…the multi-layer snow scheme provides a stronger insulating effect, simulating significantly warmer soil temperature from snow cover onset through March, when air is colder than the land surface (Fig. 1c). In GloSea6, the colder surface air temperature during the snow freezing season is attributable to the energy interception between the air and the ground (Fig. 1d). The snow insulating effect also contributes to the higher air temperature during the peak snow season, limiting transfer of heat from air to soil due to the enhanced insulation by the multi-layer snowpack."**

[Figure]

Figure2a shows that GloSea6 provides more surface soil moisture in mid-to-high latitudes of the northern Hemisphere, in addition to the positive evapotranspiration-precipitation feedbacks suggested by the authors (line 355), is it possible that the update in atmospheric physics in GloSea6 rather than the update of snowpack scheme from single layer to multi-layer results in more precipitation than GloSea5 in these regions (Figure 6a,b)?

➜ *To illustrate the physical sequence between land surface variables, we look into the climatology of 25-day running mean (removing high-frequency noises) time series of surface albedo and water budget terms in the runs initiated at each year of 1ˢᵗ March (Fig. 1e). On March 21, the*

*surface albedo of GloSea6 becomes larger than that of GloSea5, and the increase in soil moisture due to the decrease in latent heat flux appears about 3 days later, and it can be confirmed that precipitation increases one day after the increase in soil moisture. Therefore, it can be confirmed that the change in snow melting due to the improvement in snow scheme sequentially affects other variables. This further description is added in Lines 287-291*

***"To illustrate the physical sequence between land surface variables by the realization of snow physics, the time series of major water budget variables is compared between both simulations (Fig. 1e). The surface albedo of GloSea6 becomes larger than that of GloSea5 at the end of March, which results in increased soil moisture about 3 days after. The increase in soil moisture resulting from the reduction in latent heat flux, with a subsequent rise in precipitation begins after the soil moisture increase."***

➔ *Moreover, to demonstrate the causality of the positive evapotranspiration-precipitation feedbacks, lead-lag correlation for the time series of the difference between GloSea5 and GloSea6 for surface soil moisture and precipitation is conducted (Fig. 1f). The highest lead-lag correlation coefficient is observed at +1 lead-lag day with statistically significant at a 99% confidence level. Therefore, the result demonstrates the abundant soil moisture over the mid-latitude areas in GloSea6 enable increasing precipitation. Of course, whether this is a realistic response for a global model with parameterized convection is a separate matter. The description is added in Lines 292-294*

***"The lead-lag correlation between soil moisture and precipitation shows statistically significant values at 0 and +1 lead-lag day and the 1-day lagged value is the highest (Fig. 1f). In other words, the increased soil moisture in mid-latitude regions due to snowmelt likely leads precipitation based on the positive evapotranspiration-precipitation feedback."***

[Figure]

[Figure]

How to explain the deterioration of both bias and RMSE of Tmax and precipitation in GloSea6 with improved snow scheme in northeastern Eurasian continent (Figures 4e, 5f, 6f)?

➔ *In northeastern Eurasia, low Tmax and wet precipitation biases are observed in GloSea6. Although the positive bias of precipitation exists, as abundant soil moisture supplies enough moisture triggering convective rainfall, it does not significantly increase the RMSE of the modeled precipitation, however. On the other hand, the multi-layer snowpack has a cooling effect compared to the past zero-layer snow scheme, which further aggravates the cold bias observed in Glosea5 and significantly increases its RMSE. Thus, we look into the climatological seasonal cycle of Tmax over northeastern Eurasia (100-120E, 50-65N) (figure below). Comparing the observed seasonal cycle, GloSea5 and GloSea6 commonly represent systematic cold bias which is attributed by the initial condition problem. In particular, the colder initial states prescribed in GloSea6 exacerbate the cold bias. The description is added in Lines 374-376*

*"However, some errors are aggravated in GloSea6. For instance, in northeastern Eurasia, Tmax RMSE is significantly increased by an exacerbated cold bias, which is related to a cold bias in initial condition (not shown). The multi-layer snowpack reinforces this bias in GloSea6."*

[Figure]

Minor points:

Line 14 of the abstract, "permafrost extent" is not addressed in this study.

➔ *The permafrost extent is not covered in this study. It is not expressly quantified based on our research results, so that it is removed in the revised text and revised in Lines 13-15*

*"In GloSea6, the snow melting season shifts two weeks later, delaying the onset of evaporation in the spring season. This slows soil moisture drying, resulting in the improvement in its climatology and memory."*

Line 114, "single-layer snow scheme allows the surface layer of the atmosphere to directly access the heat in the soil" is not true when the snowpack is thick.

➔ *GL6, which is the LSM used in GloSea5, employs zero-layer scheme in which a single thermal store was used for snow and the first soil level, and an insulating factor was applied to represent the lower thermal conductivity of snow. The snow scheme itself included no representation of the thermal evolution of the snowpack. Because the past snow scheme has the single thermal store, we express it as "single-layer", but it has clearly led to some confusion. Thus, "single-layer" is replaced by "zero-layer" throughout the manuscript to avoid the confusion. Further description is added in Lines 114-117*

*"In GloSea5, a zero-layer snow scheme permitted direct heat exchange between the surface layer of the atmosphere and the soil, utilizing a single thermal store for both the snow and the uppermost soil layer, with an insulating factor to account for the reduced thermal conductivity of snow. This scheme lacked a dynamic representation of snowpack evolution with the inadequate depiction of the snowpack's insulating properties."*

Line 395, "(f)" should be "(c)".

➔ *Thanks for pointing out this typo. We correct it in the caption of Fig. 8.*

Line 412, "winch" should be "which".

➔ *It is also corrected in the updated manuscript.*

The title of this paper can be modified to be more appropriate for its content.

➔ *This study mainly demonstrates the improvement of model's climatological fidelity with the realization of land-atmosphere interactions by implementing multi-layer snow scheme. Based on reviewer's suggestion, we modify the title of this paper to* **"Improving land-atmosphere coupling in seasonal forecast system by implementing a multi-layer snow scheme"**.

---

## Referee Report (RR1)

The manuscript "Improving land-atmosphere coupling in seasonal forecast system by implementing a multi-layer snow scheme" evaluated the retrospective seasonal forecast performance of the Global Seasonal Forecast System (GloSea) version 5 (GloSea5) and version 6 (GloSea6) over a 24-year period (1993-2016) and tried attributing the improving retrospective seasonal forecasts in GloSea6 during winter and snow melting seasons to the implementation of multi-layer snow scheme in GloSea6. The comparison results indicated that the snow melting season shifts two weeks later in GloSea6, consequently improving the simulations of soil moisture in its climatology and memory, surface temperature, and land-atmosphere coupling regime as well. The authors thought that the subsequent improvements in surface temperature and precipitation over snow-covered regions were resulted from the implementation of multi-layer snow scheme in GloSea6. This probably holds when other model physics are same in GloSea5 and GloSea6. Therefore, the authors should find other ways to analyze the results. Here are some specific comments:

1.  From Table 1, we could see that only the model resolution in two versions of GloSea are same. The model coupler and model physics are all updated in GloSea6. The surface air temperature and precipitation are impacted not only by the local effect but also the nonlocal effect. For a coupled system, the physics interact with each other. Therefore, it is It is unrealistic to talk about only one physical process while ignoring the influence of other physical processes.

2.  From Table 3 in Kim et al. (2021), compare with GL6.0 in GloSea5, the GL8.0 in GloSea6 added the multi-layer snow scheme, improved the land surface albedo physics and modified the atmospheric rain fractions. Temporarily leaving aside the influences of modification in land surface albedo, how does the modification in partition between rain and snow affect the snow characteristics? Moreover, in around Line 120, how could the authors distinguish the albedo changes from the multi-layer snowpack or improved surface albedo physics?

3.  In around Line 265, "The multi-layer snowpack also extends the area of snow cover, which leads to the increased surface albedo, where increasing snow

amount leads to an increase of surface albedo at the land surface about 10 days later (SFs. 1a, b)". The surface albedo was closely connected with the snow cover fraction over the snow-covered regions. However, the more snow amounts couldn't always mean larger snow cover area. It would be good to compare the snow cover fraction and snow albedo in two versions. Otherwise, how to explain the similar snow amount in two versions from Oct to Jan but the different surface albedo?

4. In Line 25-30: "As the memory of initial land conditions can extend out to approximately 2 months, the importance of realistic land surface initialization in determining skill of the subseasonal forecast is paramount (Koster et al., 2011; Guo et al., 2011; Seo et al., 2019)." However, the two versions used different land initial conditions. How to exclude this effect from the effect of multi-layer snowpack?

5. Around Line 190, "reanalysis-based precipitation dataset with are available for 1979–present". Not clear.

6. The label bar in Fig.8 was missed. It's hard to follow the left-bottom-corner sub-figure.

---

## Author Response (AR3)

*Dear Prof. Jinkyu Hong,*

*Many thanks for handling the review process for our manuscript. The time and effort devoted to our manuscript by you and the reviewers are very much appreciated.*

*We have revised the manuscript carefully according to the reviewers' comments and suggestions. In the following, we provide a point-by-point response. The original reviewer comments are in black regular font. Our responses are shown in blue italic font. Quotes from the revised paper are shown in blue bold-face font. The edits are highlighted in the marked version of revised manuscript with yellow (reviewer #1), green (reviewer #2) and aqua (reviewer #4), but it may be marked in a different color, if a revision has been made based on other reviewers' comments.*

REVIEWER COMMENTS

**Reviewer #1**:

The manuscript, Improving land-atmosphere coupling in a seasonal forecast system by implementing a multi-layer snow scheme, fails to meet the standards required for publication in Geoscientific Model Development due to multiple critical issues. The title is unclear, as the manuscript does not clarify whether the multi-layer snow scheme was developed by the authors or implemented into the GloSea model by authors. Or authors here just to access its impacts. The introduction lacks detailed explanations of the mechanisms by which the multi-layer snow scheme addresses biases and fails to provide appropriate references to support the claims made. The data section is incomplete, providing insufficient detail on the differences between the single-layer and multi-layer snow schemes in GloSea5 and GloSea6 and their origins or physical basis. As model development paper, the origins, implementations, and development history of this 'multi-layer' snow scheme are the must have parts. After reading the manuscript and the responds to the reviewers, I still can't get which multi-layer snow scheme is discussed in this study. Is there only one parameterization about multi-layer snow in land model community? What is the iteration of this kind of parameterizations?

The methodology is unclear, particularly, the lack of offline simulations makes it impossible to isolate the impact of the multi-layer snow scheme from other model changes, which significantly undermines the validity of the conclusions. Offline land model simulations, as demonstrated in studies like Arduini et al. (2019), could provide more robust insights into the impact of the snow schemes. Offline simulations for GloSea5 and GloSea6, even if not for long-term historical runs, would add significant value. The results section is weakened by inconsistent comparisons—such as snow water equivalent versus snow cover—and unsupported claims regarding the improvements attributed to the multi-layer snow scheme. Importantly, authors haven't ruled out whether other changes in the atmospheric model could also influence the mid- to high-latitude regions.

Overall, the manuscript fails to provide the rigor and clarity required for a model development paper. To address these issues, the authors must (1) clarify what version of multi-layer snow scheme was discussed in this study, (2) provide detailed references and explanations for the scheme's physical basis, (3) conduct offline simulations to isolate the snow scheme's impacts, and (4) ensure consistent and meaningful comparisons of variables. Without these major revisions, the manuscript does not meet the publication standards of GMD.

> ➔ *We appreciate the time and effort the reviewer has taken to evaluate our manuscript, "Improving land-atmosphere coupling in a seasonal forecast system by implementing a multi-layer snow scheme". Your comments greatly helped us identify and address several important limitations in the original manuscript during the revision process. We have made a concerted effort to incorporate your feedback, particularly focusing on four key aspects reviewer pointed out.*

*First, we have expanded the description of the land surface model version used in the study, along with how snow is represented within the model. While a formal versioning system does not exist for the snow scheme itself, we have added detailed explanations of the scheme's configuration and behavior in Section 2 to clarify its implementation.*

*Second, we have included additional references that provide the physical basis of the multi-layer snow scheme. Relevant details from these studies have been incorporated into Introduction and Model description sections. We enhance the introduction by incorporating a more detailed explanation of how multi-layer snow schemes influence land-atmosphere coupling, particularly in addressing biases in snow representation. We will also add references to relevant studies that demonstrate these mechanisms and provide a more comprehensive background on the existing literature. Additionally, detailed information on the model used in this study and the key differences between the two model configurations has been provided in Section 2. In particular, we have elaborated on the implementation of the multi-layer snowpack scheme and the update of surface albedo. Additionally, within the section describing the JULES offline experiments, we have included a description of the snow layer structure, specifying the depths at which snow layers are formed in the multi-layer snowpack scheme.*

*Third, in order to isolate the effects of atmospheric forcing and better assess the impact of the snow scheme, we have conducted new offline land surface model experiments using JULES. Two experiments were performed under identical conditions, differing only in whether a single layer or multi-layer snow scheme was applied. The results show that the differences observed between GloSea5 and GloSea6 are reproduced in the offline simulations, confirming the influence of snow scheme changes. When compared to a state-of-the-art reanalysis product known for its high snow simulation accuracy, the multi-layer snow scheme demonstrates improved performance. In addition, we compared the offline results with those from the fully coupled forecast system to examine how snow-related land surface changes interact with the atmosphere when the models are coupled.*

*Fourth, we performed additional analyses on key land surface variables to provide a more integrated understanding of changes in the surface energy and water balance associated with the snow scheme. Furthermore, in response to your concern regarding potential misinterpretation of snow impacts over regions where snow is not a dominant factor, we revised the scope of our analysis. Specifically, we replaced the original global-scale evaluations with a focused assessment over snow-affected regions (mid- and high-latitude areas of the Northern Hemisphere) and removed interpretations related to other regions.*

*Once again, we sincerely thank the reviewer for the valuable feedback, which has significantly improved the clarity, rigor, and focus of this study. Please note that this paper has been submitted to the Model Evaluation section of GMD journal. Therefore, it primarily focuses on evaluating the land-atmosphere interaction simulated in GloSea5 and GloSea6 models, rather than on the model development*

Title
The title, "Improving land-atmosphere coupling in a seasonal forecast system by implementing a multi-layer snow scheme", raises questions about its accuracy. Did the authors implement this scheme into the GloSea model, or did they develop the multi-layer snow scheme themselves? If not clarified, the title feels inappropriate and somewhat misleading.

➔ *Although we did not personally implement the multi-layer snow scheme into the GloSea6 model, our study focuses on a detailed evaluation of land-atmosphere interactions by comparing model*

*simulations with and without the multi-layer snow scheme. Given this emphasis on assessing model performance and its impact on coupled land-atmosphere processes, we believe this study is well-suited for the **"Model Evaluation"** section of GMD journal and its title is also appropriate for this research.*

1. Introduction

Line 50: What does "coupling strength" mean in this context? Is there a specific metric to quantify this "coupling strength"? Please clarify.

➔ *We describe a previous study (Xu and Dirmeyer, 2011) on the coupling strength between snow cover and near-surface atmospheric variables. To calculate the snow-atmosphere coupling strength, we used a coupling index quantifying the agreement of members of an ensemble forecast (Koster et al., 2006). It is now specified in the main text in Lines 51-53.*

*"the coupling strength of snow cover to near-surface atmospheric variables, as measured by the phase similarity of members of an ensemble forecast induced by specifying identical land surface conditions (Koster et al., 2006), …"*

● *Koster, R. D., Sud, Y., Guo, Z., Dirmeyer, P. A., Bonan, G., Oleson, K. W., Chan, E., Verseghy, D., Cox, P., and Davies, H.: GLACE: the global land–atmosphere coupling experiment. Part I: overview, Journal of Hydrometeorology, 7, 590-610, 2006.*

Line 56: In the introduction, it would be helpful to provide clear descriptions of the "warm and cold biases during winter and snowmelt seasons" caused by the absence of a multi-layer snowpack scheme. Since these biases are a major focus of the results section, a detailed explanation of their underlying mechanisms would strengthen the introduction.

➔ *During the winter season, implementation of a multi-layer snow scheme reduces the efficiency of cold air penetration to the surface due to enhanced insulative properties of the snow. This leads to a warmer surface temperature compared to the single-layer snow scheme, which cannot simulate a vertical temperature gradient in the snow. During the snow melting season, as atmospheric temperatures rise, energy transfer to the underlying soil in a multi-layer snow scheme becomes less effective than in the single-layer scheme, resulting in a delayed snowmelt period. Consequently, the simulated temperatures during the snowmelt season are lower than those produced by the single-layer snow scheme. This detail is added in Lines 59-63.*

*"The snowpack insulates the land surface, inhibiting energy exchange between the land surface and the atmosphere. Consequently, a single layer snowpack scheme typically leads to cold and warm biases during winter and snow melting seasons, respectively. Because a single-layer scheme cannot simulate a vertical temperature gradient within the snowpack, it transmits surface temperature changes directly to the soil below, enhancing the efficiency of energy exchange."*

Line 59: What does "Noah-MP" refer to?

➔ *To clarify the notation of "Noah-MP", its full name is added in the text with **"Noah-Multiparameterization (Noah-MP)"**.*

Line 61: Is the "layered snowpack scheme" mentioned here the same as the "multi-layer scheme"?

➔ *The reviewer's understanding is correct. The text has been revised in Line 66, to make it clearer to reduce confusion.*

*"Noah-MP adopts the multi-layer snowpack scheme."*

Line 64: Please clarify the transition between Noah-MP and JULES. Are the authors providing examples of models using multi-layer schemes? Do these models employ the same "multi-layer scheme"? How many different multi-layer schemes exist within the land model community? The citation of Walters et al. (2017) is insufficient, as it is an overview paper on the JULES model rather than a specific reference for the multi-layer scheme.

➔ *The description of Noah LSM and Noah-MP written in the Introduction section now also presents an example of introducing a multi-layer snowpack scheme from a single-layer snowpack to improve the land surface model. Thus, it is now clearly written that Noah LSM uses a single-layer snowpack scheme in Lines 64-66.*

*"Noah-Multiparameterization (Noah-MP) LSM represents the latest iteration of Noah LSM, a land component widely implemented with a single-layer snowpack in various regional and global operational forecast models."*

*Regarding a specific reference for the multi-layer snow scheme in the JULES LSM, we have added Burket et al., (2013) to demonstrate the improvement in the simulation of soil temperature and permafrost extent using LSM offline simulations.*

● *Burke, E. J., Dankers, R., Jones, C. D., and Wiltshire, A. J.: A retrospective analysis of pan Arctic permafrost using the JULES land surface model, Climate Dynamics, 41, 1025-1038, 2013.*

Line 69: The names of the 13 S2S models or the study that characterizes these models are missing. Please provide a clear citation.

➔ *To clarify the description of the 13 S2S models mentioned in this manuscript, it is noted that they are models participating in the S2S prediction project in Lines 81-87.*

*"For instance, among 13 operational models participating in sub-seasonal to seasonal (S2S) prediction project (Vitart et al., 2017; Vitart et al., 2025), only three—BoM (POAMA P24), CNR-ISAC (GLOBO), and NCEP (CFSv2)—employ a single-layer snowpack scheme, whereas the remaining ten models, including those developed by CMA (BCC-CPS-S2Sv2), CNRM (CNRM-CM 6.1), CPTEC (BAM-1.2), ECCC (GEPS8), ECMWF (CY49R1), HMCR (RUMS), IAP-CAS (CAS-FGOALS-f2-V1.4), JMA (CPS3), KMA (GloSea6-GC3.2), and UKMO (GloSea6), now used multi-layer snowpack schemes. Despite this broad adoption, the impact of multi-layer snow schemes on S2S forecasts remains insufficiently explored and understood."*

● *Vitart, F., Ardilouze, C., Bonet, A., Brookshaw, A., Chen, M., Codorean, C., Déqué, M., Ferranti, L., Fucile, E., and Fuentes, M.: The subseasonal to seasonal (S2S) prediction project database, Bulletin of the American Meteorological Society, 98, 163-173, 2017.*

● *Vitart, F., Robertson, A., Brookshaw, A., Caltabiano, N., Coelho, C., de Coning, E., Dirmeyer, P., Domeisen, D., Hirons, L., and Kim, H.: The WWRP/WCRP S2S project and its achievements, Bulletin of the American Meteorological Society, 2025.*

Line 70: The statement, "Hence, this study conducts a comparative analysis between GloSea5 (single-layer snowpack) and GloSea6 (multi-layer snowpack)," is misleading. As mentioned in the "Data" section, GloSea6 involves multiple changes, with snowpack treatment being just one of them. This distinction should be made clear early on.

➔ *We modified the sentence in Lines 87-89, because we have added JULES offline simulations for the comparison between single layer and multi-layer snowpack schemes.*

**"Hence, this study conducts a comparative analysis between single layer and multi-layer snowpack in the JULES LSM, as well as the fully coupled forecast systems GloSea5 and GloSea6"**

Line 73: The primary and secondary objectives of the study are vague. Why does the primary objective receive only a single sentence, while the secondary objective is elaborated in more detail? Which of these objectives is the study's main focus?

➔ *The imbalance between primary and secondary objective, pointed by the reviewer, has been rectified by separating the investigation on the climatological model performance from diagnosing the model fidelity in simulating land-atmosphere interactions. The text has been edited in Lines 90-92 and 95-96.*

**"The primary objective of this study is to assess the seasonal cycle of snow and land surface variables throughout the snow-covered period and evaluate the model's capability to replicate the mean climatology of key land surface and near-surface atmospheric variables"**

**"Furthermore, the model fidelity in the simulation of land-atmosphere interactions, corresponding to water- and energy-limited processes, is diagnosed to identify the realism of land coupling regimes by implementing the advanced snowpack scheme."**

In general, the introduction needs significant improvement. It lacks references detailing the multi-layer snowpack scheme and its physical or mathematical basis. Although it is possible that the multi-layer scheme performs better than the single-layer scheme, the mechanisms remain unclear. While summarizing the development of land models and snowpack treatments is challenging, Geoscientific Model Development (GMD) requires a more thorough and rigorous introduction to meet its high standards.

➔ *In the introduction section, we have added a description of the characteristics of the multi-layer snowpack scheme and its impact on land surface processes. In particular, we have written more detailed information on the multi-layer snowpack scheme applied to the JULES model, used in this study, to emphasize the reason for using the multi-layer snowpack scheme and the significance of this study. It is added in Lines 71-77.*

**"It also dynamically adjusts the number of snow layers, with each layer having prognostic variables for temperature, density, grain size, and both liquid and solid water content (Best et al., 2011). Unlike the simpler single layer snow model, which treats snow as an adaptation of the top-soil layer, the multi-layer scheme accounts for independent snow layer evolution and the impact of snow aging on albedo through simulated grain size changes. By explicitly simulating snow insulation effects and meltwater percolation, this scheme better captures seasonal snow variability and its influence on soil thermal regimes, including surface cooling during winter, delayed ground thaw in spring, and subsurface heat retention in summer."**

- *Best, M., Pryor, M., Clark, D., Rooney, G., Essery, R., Ménard, C., Edwards, J., Hendry, M., Porson, A., and Gedney, N.: The Joint UK Land Environment Simulator (JULES), model description–Part 1: energy and water fluxes, Geoscientific Model Development, 4, 677-699, 2011.*

2. Data

Line 83: Using "Data" as the section title is not ideal. A more precise title would be beneficial.

➔ *To specify the section #2, we have modified the section title to **"Model Description and Data"**.*

Line 90: Additional clarification about JULES and GL would help readers understand the model structure. Are these the same land model?

➔ *To clarify the notation of JULES and GL, we explicitly note that GL8.0 uses JULES version 5.6 as the name specified in the coupled forecast system. It is added in Lines 184-185.*

**"we conduct two sets of LSM offline experiments using GL8.0 (representing a specific configuration of JULES version 5.6 within the coupled system)"**

Line 100: The citation of Kim et al. (2021) is inaccurate. The paper focuses on atmospheric improvements in GloSea6 and does not provide an overview of "all model components." Please revise this characterization.

*Based on reviewer's suggestion, we replaced the reference to provide overviews of the core components of both GloSea5 (Williams et al., 2015) and GloSea6 (Williams et al., 2018) models.*

- *Williams, K., Harris, C., Bodas-Salcedo, A., Camp, J., Comer, R., Copsey, D., Fereday, D., Graham, T., Hill, R., and Hinton, T.: The Met Office global coupled model 2.0 (GC2) configuration, Geoscientific Model Development, 8, 1509-1524, 2015.*

- *Williams, K., Copsey, D., Blockley, E., Bodas-Salcedo, A., Calvert, D., Comer, R., Davis, P., Graham, T., Hewitt, H., and Hill, R.: The Met Office global coupled model 3.0 and 3.1 (GC3. 0 and GC3. 1) configurations, Journal of Advances in Modeling Earth Systems, 10, 357-380, 2018.*

Lines 113–115: There is confusion regarding the single-layer snow scheme in GloSea5. The authors state that it has constant thermal conductivity and density but later mention adjustments for layer thickness. Is the thermal conductivity constant or not in GloSea5? Additionally, a reference for the origin of this single-layer scheme is necessary.

➔ *We apologize for the confusion. In the single layer snow scheme, the snow and the uppermost soil layer are treated as a single thermal store, and the increased snow depth leads to a reduction in the effective thermal conductivity. However, this reduction is not a dynamic change in the snow's intrinsic thermal properties, but rather an adjustment to account for the insulating effect of the snow. The description about the single layer snowpack scheme is edited in Lines 134-138 along with a reference to the origin of the snow scheme.*

**"GloSea5 has a single layer snow scheme, in which snow is assigned a constant thermal conductivity and density, allowing direct heat exchange between the surface atmosphere and**

*the soil (Best et al., 2011). It combines the snow and the uppermost soil layer into a single thermal store, with the increased snow depth leading to a reduction in the effective thermal conductivity. This reduction is not a dynamic representation of the intrinsic properties of snow but rather an adjustment to account for the insulating effect of the snow."*

- *Best, M., Pryor, M., Clark, D., Rooney, G., Essery, R., Ménard, C., Edwards, J., Hendry, M., Porson, A., and Gedney, N.: The Joint UK Land Environment Simulator (JULES), model description–Part 1: energy and water fluxes, Geoscientific Model Development, 4, 677-699, 2011.*

Line 119: Walters et al. (2017) does not discuss snowpack treatment or the multi-layer snow scheme. This citation is inappropriate.

➔ *We made a mistake citing a 2019 paper by the same author. It is corrected in the revised manuscript.*

The data section should include a clear description of the snowpack treatments in GloSea5 and GloSea6. For the multi-layer snow scheme, its origin, physical or mathematical improvements over the previous treatment, and whether it was developed by the authors should be explicitly stated. These details are critical for a model development or assessment paper.

➔ *Thank you for your thorough review of the paper. We have revised it to effectively convey the research by reflecting the reviewer's comments as much as possible. In particular, in order to diagnose the impact corresponding to implementation of different snow schemes, we have performed additional offline LSM experiments to isolate the impact of advanced snow physics on the simulation of land variables. The fully coupled model takes into account the interaction between the land and atmosphere. Thus, the result of comparisons between GloSea5 and GloSea6 can be more firmly attributed to either the interactions between land and atmospheric models or the changes in the land model itself.*

3. Methodology

Lines 209–216: The authors state that "a single-member run is used in this study" but later mention "ensemble mean values" for bias analysis. The term "ensemble" is used inconsistently throughout the manuscript. Please clarify whether the results are based on a single-member run or ensemble simulations.

➔ *We apologize for not being clear about which results are single member results, and which are ensemble results. In the revised manuscript, we try to clarify this confusion in Lines 281-282.*

*"this study uses a single member run only for analyzing the climatology of the seasonal cycle (Fig. 2), since 24 yearly samples are sufficient."*

In coupled ensemble simulations, it is challenging to identify which model components drive improvements in surface temperature, soil moisture, and other variables. Offline land model simulations, as demonstrated in studies like Arduini et al. (2019), could provide more robust insights into the impact of the snow scheme. Offline simulations for GloSea5 and GloSea6, even if not for long-term historical runs, would add significant value. While coupled model analysis is useful, comparing offline and coupled results would greatly strengthen the study.

➔ *We fully agree with the reviewer's comments. It is important to understand the changes in*

*surface variables due to the use of the multi-layer snowpack scheme by running the LSM offline, where the atmospheric influence is removed. By comparing the offline simulations with coupled model runs, we can understand the differences in the forecast models when coupled with atmospheric components. Therefore, we have added the results for offline LSM model simulations in the revised manuscript.*

4. Results

Line 277 and Figure 1: The comparison between GloSea5 and GloSea6 uses snow water equivalent and ERA5 snow cover percentage—two different variables. This mismatch should be clarified. Additionally, it is premature to conclude that GloSea6 snow water equivalent is superior to GloSea5 based on ERA5 snow coverage without further explanation or an "apples-to-apples" comparison.

➔ *In order to directly compare snow water equivalent, JRA-3Q reanalysis is used as reference data. In a previous study (Orsolini et al, 2019), the performance of JRA-55 reanalysis data was found to be better when comparing snow cover and depth among other reanalysis data with satellite-based and in situ datasets. The updated version of JRA-55 called JRA-3Q, which further improves the problems that affect the snow simulation, is now used to validate the climatology of seasonal cycle of snow water equivalent. Its description is added in Lines 216-225 and 403-404.*

**"This study uses Japanese Reanalysis for Three Quarters of a Century (JRA-3Q; Kosaka et al., 2024) as a reference for snow water equivalent to diagnose the modelled snow. It employs an offline version of the Simple Biosphere (SIB) model (Sellers et al., 1986). Compared to the satellite-based and in situ datasets, the snow cover and depth are accurately described in its predecessor, the Japanese 55-year Reanalysis (JRA-55) (Orsolini et al., 2019). JRA-3Q incorporates daily snow depth data from the Special Sensor Microwave/Imager (SSM/I), the Special Sensor Microwave Imager Sounder (SSMIS), and in situ measurements using a univariate two-dimensional optimal interpolation (OI) approach. Although this procedure is comparable to that used in JRA-55 (Kobayashi et al., 2015), two issues—unrealistic analysis near coastal areas and unintended increments caused by satellite data biases—have been resolved in JRA-3Q. Additionally, JRA-3Q employs the multi-layer snowpack scheme whereas JRA-55 used a single layer snowpack scheme. JRA-3Q has a horizontal resolution of 0.375˚ and 3-hourly temporal resolution."**

**"The result resembles the snow dissipation represented by JRA-3Q, particularly in the run initiated on 1st April."**

● *Kosaka, Y., Kobayashi, S., Harada, Y., Kobayashi, C., Naoe, H., Yoshimoto, K., Harada, M., Goto, N., Chiba, J., and Miyaoka, K.: The JRA-3Q reanalysis, Journal of the Meteorological Society of Japan. Ser. II, 102, 49-109, 2024.*

● *Sellers, P., Mintz, Y., Sud, Y. e. a., and Dalcher, A.: A simple biosphere model (SiB) for use within general circulation models, Journal of Atmospheric Sciences, 43, 505-531, 1986.*

● *Orsolini, Y., Wegmann, M., Dutra, E., Liu, B., Balsamo, G., Yang, K., de Rosnay, P., Zhu, C., Wang, W., and Senan, R.: Evaluation of snow depth and snow cover over the Tibetan Plateau in global reanalyses using in situ and satellite remote sensing observations, The Cryosphere, 13, 2221-2239, 2019.*

● *Kobayashi, S., Ota, Y., Harada, Y., Ebita, A., Moriya, M., Onoda, H., Onogi, K., Kamahori, H., Kobayashi, C., and Endo, H.: The JRA-55 reanalysis: General specifications and basic characteristics, Journal of the Meteorological Society of Japan. Ser. II, 93, 5-48, 2015.*

Line 278: The discussion of albedo differences between GloSea5 and GloSea6 is significant and should be shown in the main part of the manuscript but not supplementary material. The larger initial albedo difference shown in SF.3b should be explained, particularly as it diminishes over time. Is it related to the surface albedo treatment changes in the land model between GloSea5 and GloSea6.

➔ *We agree with the reviewer's comment that, in addition to the changes in the snow scheme, the modifications to surface albedo in the GloSea6 model are important enough to be discussed in the main text (subsection 2.1) when comparing GloSea5 and GloSea6. Therefore, we have incorporated the figure originally presented in the supplementary material into Figure 2 and added a corresponding explanation in the main text. The implementation of the multi-layer snow scheme primarily affects surface albedo during the snow season when snow is present, while the modification of surface albedo in GloSea6 affects both snow-covered and snow-free seasons. However, when snow is absent, the difference between GloSea5 and GloSea6 appears to be minimal. This suggests that, although the albedo was updated, its impact is not substantial in the absence of snow, and therefore we interpret the difference between GloSea5 and GloSea6 outside this season as being primarily related to the impact of the multi-layer snowpack scheme. We have added this explanation to the revised manuscript in Lines 156-160.*

*"The shift from bare soil to vegetated surfaces decreases surface albedo (Fig. 2e), as the vegetation can penetrate snow cover during the winter season (SF. 2a). Therefore, the surface albedo differences observed during the snow-covered season can be attributed to amendments in land surface type classification, whereas the albedo differences during the snow-free period are understood to result from the incorporation of wavelength-dependent calculations in the surface albedo scheme."*

Line 286: The conclusion that GloSea6's improvements are due to the multi-layer snow scheme is unconvincing, especially since the largest differences occur in October and November when snow cover is minimal. Could these differences be attributed to changes in precipitation or large-scale circulation strength?

➔ *We appreciate the reviewer's comment highlighting the concern that the largest differences in soil moisture occur in October, when snow cover is minimal, potentially weakening the attribution to the multi-layer snow scheme. We agree that during this period, other factors could also contribute to model discrepancies. To clarify, we have revised the manuscript (Lines 408– 414) to better distinguish the contributions from different physical drivers across seasons. Specifically, we now acknowledge that the wetter soil moisture state simulated in GloSea5 during October is likely attributable to a positive precipitation bias, rather than snow-related processes.*

*"Because the snowpack serves as a barrier to energy and water exchange between the land and the atmosphere, in the single layer snowpack, colder soil temperatures lead to a model drift toward wetter conditions during the snow-covered season, consistent with the results from the JULES LSM simulations (cf. Fig. 1e,g), and the early onset of evaporation manifests the physical process of drying out SM during snow melting season. Wetter soil moisture is simulated in GloSea5 during October, when snow cover is minimal, which is attributed to a positive precipitation bias (not shown). Thus, the implementation of the multi-layer snowpack results in the climatologically dryer and wetter SM, respectively, preceding (November–March) and following (April–June) the onset of snowmelt."*

*Our conclusion regarding the role of the multi-layer snowpack scheme is focused on the snow-*

*covered and melting seasons (November–June). Notably, the soil moisture evolution in GloSea6 shows delayed drying following snowmelt, which aligns with the insulating effect and melt timing represented by the multi-layer snow scheme. We also emphasize that in the JULES offline experiments where other model differences are excluded.*

Line 309: A correlation coefficient of ~0.35 explains only ~10% of the variance between soil moisture (SM) and precipitation (PR). Correlation does not imply causation. The authors need to rule out other potential factors influencing precipitation and SM before concluding that SM changes driven by the multi-layer snowpack explain precipitation differences.

➜ *If the correlation coefficient between soil moisture (SM) and precipitation (PR) in an actual model is around 0.35, as the reviewer pointed out, the relationship can be considered relatively weak—statistically significant but explaining only a small portion of the variance. However, in this study, we do not calculate a simple correlation between the two variables. Instead, we compute the correlation based on the differences in SM and PR between GloSea5 and GloSea6, focusing on how the relationship between the two variables changes due to differences in model configuration. From this perspective, the correlation values are not negligible. Moreover, since simple correlation does not imply causation, we use time-lag analysis to provide an indirect assessment of causal relationships between variables. Although the figure caption explains that the correlations are based on the time series of differences between GloSea5 and GloSea6, this may have caused confusion because it was not clearly reflected in the figure itself. In the revised manuscript, we have clarified this by updating the y-axis label in Figure 2h.*

➜ *To support the causality between evaporative fraction and precipitation, we replace the time-lagged correlation between both variables with the results from a Granger causality test. This is a statistical principle to identify the potential dependence of evaporative fraction and precipitation. We have added the description of the granger causality in evaporation-precipitation feedback in **subsection 3.2** and have replaced corresponding **Figure 8** along with its description in the main text.*

[Figure]

The change of precipitation can cause the change of the snow, which leads to albedo difference. The driver of the precipitation could be the snowpack treatment, but also could be the convection, aerosol, and cloud physics changes between GloSea5 and GloSea6.

➜ *Thank you for raising the point regarding the influence of winter precipitation on snow-related changes. As the reviewer's suggestion is entirely valid, we examined the seasonal cycle of precipitation simulated by GloSea5 and GloSea6 in bottom figure. Overall, GloSea6 simulates slightly more precipitation during the winter season; however, the difference between the two models during winter and spring is only about 0.1 mm. When this is compared with the*

*differences in snow water equivalent (SWE) between the two experiments, the impact of precipitation appears limited—particularly in early winter (November–December), where the SWE difference is relatively small. In spring, the SWE difference reaches approximately 6 mm between the two simulations, suggesting that the influence of precipitation on snowpack differences is likely minor relative to the changes induced by the snow scheme itself. Its description is added in Lines 394-398.*

**"Differences in winter precipitation between both models may lead to variations in snow accumulation; however, although GloSea6 generally simulates slightly higher precipitation, the magnitude of this difference is negligible compared to the difference in snow water equivalent (not shown). Therefore, the impact of precipitation on snow accumulation is not considered in this study."**

[Figure]

Line 331: For the simulated SM differences between GloSea6 and GloSea5, it is true that there are large differences over the snow frontal region. However, there are also significant differences over the Amazon rainforest in South America, where snow is rare.

➔ *To focus on snow-related impacts throughout the manuscript, the original global-scale analyses have been revised to present results specifically for mid- and high-latitude regions. Differences observed in regions that are not directly influenced by snow are likely caused by other model changes, and convective precipitation, which dominates in the tropics, is notoriously chaotic and may not be related to model changes at all. Therefore, all related contents outside snow-covered areas have been removed.*

Line 334: Why do other model changes tend to impact tropical precipitation or SM but show no clear impact on northern mid- to high latitudes? While I do not deny that the snowpack treatment change contributes, there is no evidence ruling out whether other changes could also influence the mid- to high-latitude regions.

➔ *We appreciate the reviewer's thoughtful comment. We fully agree that other model updates implemented in GloSea6—such as changes in convection, radiation, or land surface albedo— may also influence climate characteristics in the mid- to high-latitude regions through the change of the meridional circulation. Walter et al., (2019) addressed the updates in atmospheric model, alterations to the meridional circulation are confined to tropical regions. In the original*

*manuscript, our intention was not to exclude the potential impact of these changes, but to emphasize the added value of the improved snowpack physics, particularly in snow-covered regions where the multi-layer snow scheme is expected to play a dominant role. To address this concern, we remake the limitation of this study in conclusion section in Lines 640-646.*

*"However, differences between GloSea5 and GloSea6 in areas unrelated to snow (e.g., India, South Asia, and East Asia) likely result from various other factors arising from other modifications as part of the model version update. Although atmospheric updates may alter the meridional circulation by modifying atmospheric variability in the tropics, their impacts are predominantly confined to tropical regions, with limited influence over the mid- or high-latitude regions (see Fig. 14 in Walters et al., 2019). As it is not possible to fully isolate the contributions of other model components, this study focuses on the mid- and high-latitude regions of the NH to better attribute local land surface processes to improvements in snow physics."*

Line 359: The "significant improvement" in simulated SMM is unclear. A reduction from -3.7 to -3.3 days in SF.5d,e needs further explanation to justify it as a significant improvement.

➔ *We have confirmed that the soil moisture memory (SMM) simulated by GloSea6 shows improved performance compared to GloSea5 when evaluated against both reanalysis datasets and in situ observations. To confirm the statistical significance in SM memory, in the revised manuscript, the statistical significance of SMM biases in both simulations and their difference between GloSea5 and GloSea6 is tested using a Monte Carlo approach. The probability of a significant SMM is estimated by randomly generated 100 SMM samples in each observational and modelled dataset. The description of testing statistical significance is added and amended in Lines 312-320 and 493-496, respectively.*

*"Additionally, the statistical significance of SMM biases in both simulations and their difference between GloSea5 and GloSea6 is tested using a Monte Carlo approach. The probability of a significant SMM is estimated by random sampling, where randomly selected yearly JJA SM time series (92 samples) are used to create all-years JJA time series, repeatedly, to generate 100 samples in observational and modelled datasets. For testing the statistical significance of the modeled SMM biases, randomly calculated SMMs from time-filtered CCI, ERA5-Land, and GLEAM products are used to generate 300 observational samples (3 products × 100 random SMMs), which are compared to 300 and 700 random samples from GloSea5 (3 ensembles × 100 random SMMs) and GloSea6 (7 ensembles × 100 random SMMs), respectively, using a Student's t-test. The statistical significance of the SMM difference between the two model simulations is also tested with the randomly calculated 300 and 700 SMM samples."*

*"When the assessment is performed with in-situ measurements (SF. 6), an extended SMM in GloSea6, compared to GloSea5, is a better match to the observations (SFs. 6d,e). When the soil becomes wet due to the late onset of snow melting, the SM decay in response to rainfall is slow, thereby significantly increasing the SMM in mid-latitude regions (Fig. 4f)."*

Line 368: The authors claim that GloSea6 reduces the surface air temperature bias. Is this claim supported by previous studies, or is it based on the authors' analysis? Please clarify how the temperature bias data was derived and what control temperature data was used for comparison.

➔ *Before presenting a detailed evaluation of daily and sub-daily temperatures simulated by*

*GloSea5 and GloSea6, we have added an overarching conclusion at the beginning of the relevant paragraph to clearly state the main finding of this study. This statement reflects the main claim of our analysis, and no previous studies have specifically addressed this aspect. To clarify this intention and avoid potential confusion, we have revised the manuscript in Lines 504-506.*

**"Features of the surface air temperature simulation in GloSea6 during the NH warm season include reduced biases in both daily mean and sub-daily timescales across all forecast lead times (Fig. 5), which can be explained by the updated land surface physics, including changes in snow and soil processes."**

Line 371: Please clarify why decomposing Tmean into Tmax and Tmin helps identify the impacts of two major modifications in the LSM. I would appreciate further explanation on this reasoning.

➔ *As the reviewer pointed out, the original manuscript lacked a clear rationale for the sub-daily temperature analysis. To address this, we have added an explanation in the revised manuscript to clarify the motivation behind this approach. Its description is added in Lines 509-514.*

**"Both daytime and nighttime temperatures are analysed in addition to daily mean temperature to assess whether temperature changes associated with land surface processes occur preferentially during the day or night. Since many coupled land-atmosphere processes are more active during the daytime due to greater available energy (net radiation), sub-daily analysis is essential for realistically capturing their effects (Yin et al. 2023). Furthermore, relying solely on Tmean can be misleading, as it conflates errors in maximum and minimum temperatures, and thus does not necessarily reflect an overall improvement in model performance (Seo et al., 2024)."**

Line 376: If the GloSea6 simulation tends to produce more snow than GloSea5, it could lead to similar temperature reductions. However, many factors could contribute to this outcome. While the multi-layer snowpack treatment might be one factor, other potential contributors must also be considered.

➔ *We fully agree that multiple factors may influence the simulated temperature differences between GloSea5 and GloSea6, including changes beyond the snow scheme. Although the results from JULES offline experiments are included in the revised manuscript, the lack of coupling to the atmosphere hinders the response of the multi-layer snowpack to surface air temperature. However, the fact that surface cooling predominantly occurs during daytime, when land–atmosphere interactions are most active, along with the limited influence of other updates in land surface processes on surface albedo, suggests that the effect is primarily attributable to the implementation of the multi-layer snow scheme. To clarify this point, we have edited the main text in Lines 433-436 and 516-518.*

**"In summer, net radiation increases again, primarily due to a reduction in upward longwave radiation associated with surface cooling, rather than being caused by changes in surface albedo. In other words, the impact of the implementation of the multi-layer snowpack scheme is predominant rather than other modifications in land processes during the summer season."**

**"The effect of the multi-layer snow scheme on forecasting temperature is primarily surface cooling over snow frontal areas throughout the entire day (Fig. 5c), even though the temperature response is more sensitive during the daytime when land-atmosphere interactions are most active (Figs. 5f,i)."**

Line 391: In Figure 10d, values over Australia are predominantly negative, resembling those over the northern mid- to high latitudes. Therefore, it is difficult to conclude that the factors reducing biases over Australia are different from those affecting northern mid- to high latitudes.

➔ *To focus specifically on snow-related impacts, the original global-scale analyses have been refocused to only for snow-affected regions, primarily over the mid- and high latitudes. Differences in regions not directly influenced by snow are likely attributable to other causes, as mentioned previously; thus, all content related to non-snow-covered areas has been excluded from the analysis.*

Line 408: Please explain the assertion that "the improvement in the mid- and high-latitude regions of the Northern Hemisphere is likely due to the improved snow physics." In Section 2, the authors mention numerous changes to the atmospheric models used in GloSea5 and GloSea6 (Walter et al., 2017). Attributing these improvements solely to snow treatment, without considering the contributions of atmospheric changes, is not substantiated.

➔ *Unlike other land surface variables that are more directly influenced by land processes, precipitation patterns can be strongly affected by atmospheric dynamics and even ocean-atmosphere coupling. Although Figures 2h and 8 suggest a potential influence of land surface processes on precipitation, we acknowledge that this assertion may appear overly strong, as pointed out by the reviewer. Therefore, we have removed the corresponding statements from the revised manuscript to avoid over-attribution.*

5. Summary and Conclusion

Line 485: The conclusion that differences between GloSea5 and GloSea6 in areas without snow (e.g., South and East Asia, Central Africa, South America, and Australia) are likely due to other factors arising from model version updates, while differences in snow-covered areas are attributed to the snowpack treatment, is interesting but insufficiently substantiated. This hypothesis could serve as the motivation for the study but cannot be presented as its conclusion without stronger evidence.

The authors must provide clear evidence demonstrating that the factors affecting non-snow regions do not influence snow-covered regions. In a global coupled model, atmospheric physics changes, such as those in convection, clouds, and aerosols, can have wide-reaching impacts. It is possible that both atmospheric physics updates and the snowpack treatment contribute to the observed results. However, it is the authors' responsibility to isolate the specific contributions of the snowpack treatment.

I recognize that isolating the impact of land modifications in a coupled simulation is challenging. However, accessing and analyzing offline simulations would provide a more robust approach to distinguish the effects of the snowpack treatment from other model changes. This would significantly strengthen the conclusions and the overall quality of the study.

➔ *We generally agree with the reviewer's concerns. Specifically, we acknowledge the inappropriateness of attributing model differences in non-snow-affected regions to snow-related processes, as well as the difficulty of diagnosing changes in land surface variables associated with snowpack layering without supporting offline experiments. To address these issues, the revised manuscript now focuses exclusively on snow-affected regions, and the original global-domain analyses have been replaced with analyses over the mid- and high-latitude regions of the Northern Hemisphere. Interpretations and explanations related to non-snow-affected regions have been removed. In addition, we have conducted two sets of JULES offline experiments using both the single-layer and multi-layer snow schemes and added the resulting differences in land surface variables as Figure 1. A comparison between the offline*

*simulations and the fully coupled forecast system has been included in the revised manuscript to isolate land model impacts and further assess how land surface changes evolve when coupled with the atmospheric model.*

**Reviewer #2**: This paper presents a study on the seasonal evolution and climate of two models: GloSea5 and GloSea6. GloSea6 is the result of a major upgrade of GloSea5 with changes predominantly in the physics package: among a lot of details a new aerosol climatology, changes to the gas optics of the radiation scheme, changes to the cloud overlap handling, new ice optical properties, new micro-physics, changes to the gravity wave scheme, cloud top entrainment and convection closure. Also the land coupling was substantially changed, through the replacement of a single layer snow scheme by a multilayer version, a new vegetation climatology, and introduction of a wavelength dependent albedo.

This is a major model change with impact ranging from local processes to general circulation. The authors did a series of 100-day ensemble forecasts with initial conditions from October to April covering many years. The seasonal evolution of parameters related to snow is evaluated with focus on snow processes, albedo, and vegetation. Unfortunately, it is not always clear whether other processes (e.g. radiation) play a role. The tentative conclusion is nicely summarised in Fig 11, with the multi-layer snow scheme having less soil moisture in the snow season more soil moisture in summer, higher soil temperatures in winter and lower soil temperatures in summer, and 2 weeks longer snow cover. Attribution is predominantly to conductivity of the total snow layer, where the multi-layer model has a much stronger insulating effect.

The paper is a clear illustration of how difficult model development is. The main difficulty is that verification at the process level is very limited. Verification relies heavily on data sets that are at best constrained to some extent by satellite observations, and calibrated with in situ observations. Turbulent latent flux at the surface from GLEAM is a clear example. It is a good product compared to others, but it relies on surface net radiation from satellite or re-analysis and vegetation activity from satellite in clear sky conditions. Furthermore it uses the simple Priestley-Taylor formulation, which puts all the emphasis on correlation with radiation and not on atmospheric humidity. This is not a criticism, it is state of the art. The issue is that the vegetation response to environmental conditions is not well understood.

In conclusion, I recommend publication after some revision. The paper is well written, a nice set of diagnostics is presented, and the results are presented in a balanced way. The conclusions are not really firm, but the reader can decide for her/himself how to interpret the results. There are a few points, I would like the authors to address.

➔ *We appreciate the time and effort the reviewer has taken to evaluate our manuscript. Your comments greatly helped us identify and address several issues in the original manuscript. We hope we have adequately clarified our descriptions and addressed the points raised.*

Major points:

1. The paper suggests that the insulating effect cannot be properly represented with a single layer. I disagree, because the increase of thickness of a single layer will increase the insulating effect. In the implementation of GloSea5, it may not work that way, perhaps because snow is combined with the top soil layer. However, a single layer model could have been implemented such that insulation increases with the layer thickness. What a single layer cannot represent is a range of response time scales. With a thicker layer there is more inertia and slower response. For instance, the multi-layer scheme should show a much better diurnal cycle of temperature, something that is hardly discussed in the paper. Fig. 4 shows something about the diurnal cycle but all seasons are put together and therefore it is hard to see snow signals.

➔ *We appreciate the reviewer's insight regarding the potential for increased insulation in a single-layer snow model by increasing the effective thickness. Indeed, in principle, thermal inertia can*

*be enhanced by increasing thickness and modifying the properties of the snow layer. However, as described in Best et al. (2011), the single-layer scheme implemented in GloSea5 combines snow with the uppermost soil layer into a single thermal store, which simplifies the vertical structure but limits the dynamic response of the snowpack to atmospheric forcing. This approach does not allow for explicit vertical gradients in temperature, nor for the evolving stratification and metamorphism of snow layers over time.*

*On the other hand, the multi-layer snow scheme used in GloSea6 discretizes the snowpack into multiple layers with distinct thermodynamic and hydrological properties (Walters et al., 2019). This enables the representation of multiple response timescales in energy fluxes and allows for the realistic simulation of snowpack processes such as densification, meltwater percolation and refreezing, and temporal evolution of thermal conductivity. These features are essential for capturing the lagged and depth-dependent thermal behavior of snow, especially during the melt season. When the multi-layer snowpack scheme is implemented in Noah-MP land surface model, it can solve the problem of snow melting about a month early (Niu et al., 2011).*

*Regarding the impact of the multi-layer snow scheme on sub-daily temperature simulation, the implementation of the multi-layer snow scheme affects land surface processes, which in turn influence surface air temperature through land–atmosphere interactions. Thermal cycles are progressively damped with depth in the snowpack. Therefore, sub-daily temperature analysis was conducted based on the hypothesis that the impact of the multi-layer snow scheme would be more pronounced during daytime, when energy fluxes are greater and land–atmosphere coupling is otherwise stronger. This description is written in Lines 509-515.*

***"Both daytime and nighttime temperatures are analysed in addition to daily mean temperature to assess whether temperature changes associated with land surface processes occur preferentially during the day or night. Since many coupled land-atmosphere processes are typically more active during the daytime due to greater available energy (net radiation), sub-daily analysis is essential for realistically capturing their effects (Yin et al. 2023). Furthermore, relying solely on Tmean can be misleading, as it conflates errors in maximum and minimum temperatures, and thus does not necessarily reflect an overall improvement in model performance (Seo et al., 2024)."***

- *Best, M., Pryor, M., Clark, D., Rooney, G., Essery, R., Ménard, C., Edwards, J., Hendry, M., Porson, A., and Gedney, N.: The Joint UK Land Environment Simulator (JULES), model description–Part 1: energy and water fluxes, Geoscientific Model Development, 4, 677-699, 2011.*

- *Walters, D., Baran, A. J., Boutle, I., Brooks, M., Earnshaw, P., Edwards, J., Furtado, K., Hill, P., Lock, A., and Manners, J.: The Met Office Unified Model global atmosphere 7.0/7.1 and JULES global land 7.0 configurations, Geoscientific Model Development, 12, 1909-1963, 2019.*

- *Niu, G. Y., Yang, Z. L., Mitchell, K. E., Chen, F., Ek, M. B., Barlage, M., Kumar, A., Manning, K., Niyogi, D., and Rosero, E.: The community Noah land surface model with multiparameterization options (Noah-MP): 1. Model description and evaluation with local-scale measurements, Journal of Geophysical Research: Atmospheres, 116, 2011.*

- *Yin, Z., K. L. Findell, P. A. Dirmeyer, E. Shevliakova, S. Malyshev, K. Ghannam, N. Raoult, and Z. Tan, 2023: Daytime-only-mean data can enhance understanding of land-atmosphere coupling. Hydrol. Earth Sys. Sci., 27, 861-872, doi: 10.5194/hess-27-861-2023.*

2. Are there any offline simulations with forcing at say 10m above the surface from wind temperature, humidity, downward radiative fluxes and precipitation to test the snow scheme, albedo effects and vegetation changes? I know it is beyond the scope of the current paper, but something of this nature must have been done during the development of the new snow scheme. This would be extremely helpful in disentangling the impact of the main aspects that are believed to be responsible for the impact that is seen in the seasonal integrations.

➔ *We agree with the reviewer's point. To isolate the effects of atmospheric forcing and better assess the impact of the snow scheme, we have conducted offline land surface model experiments using the JULES LSM. Two experiments were performed under identical conditions, differing only in whether a single layer or multi-layer snow scheme was applied. The results show that the differences in snow-covered areas observed between GloSea5 and GloSea6 are reproduced in the offline simulations, confirming the influence of snow scheme changes. The result is included in the revised manuscript within Figure 1. When compared to a state-of-the-art reanalysis product known for its high snow simulation accuracy (JRA-3Q), the multi-layer snow scheme demonstrates improved performance. In addition, we compared the offline results with those from the fully coupled forecast system to examine how snow-related land surface changes further interact with the atmosphere when coupled.*

3. It is not clear why the snow stays 2 weeks longer on the ground with the multilevel snow scheme. It is suggested that the higher soil temperature is playing a role but I doubt it. The main source of heat for melting comes from the atmosphere. So it would be good to look at all components of the surface energy balance. A major mechanism for melt is from sensible heat flux in case of partial snow cover. Solar radiation heats the snow-free areas, which brings the air above zero. The warm air melts the snow over the snow covered fraction. This is one of the (perhaps very few) advantages of a tiled surface coupling. Perhaps the authors can comment on where the heat for snow melt is coming from.

➔ *Thank you for the insightful comment. To enhance the process-based understanding of the snowmelt mechanism, we have included results for all components of the surface energy balance. Given that the primary source of energy for snowmelt is the atmosphere, we focused on the temporal relationship between surface air temperature and snowmelt. Snow begins to melt in March, and the slower rate of melt observed in GloSea6 is associated with surface air temperature cooling, which can be attributed to evaporative cooling from increased latent heat flux. However, the drivers of the latent heat flux increase vary by season: in spring, it is primarily linked to increased net radiation, while in summer, it is driven by enhanced soil moisture. We have added this explanation to the revised manuscript and included analyses of net radiation, sensible heat flux, and latent heat flux in Figure 2 to illustrate the seasonal evolution of the surface energy balance. Its description is written in Lines 399-400, 422-425.*

**"Given that the primary source of energy for snowmelt is the atmosphere, snow melting process is tied to the variation of surface air temperature (cf. Fig. 2d)."**

**"The air temperature cooling observed from mid-March is associated with evaporative cooling driven by increased latent heat flux. During early spring, the increase in latent heat flux is primarily linked to enhanced net radiation (Fig. 2g). However, after April, the continued rise in latent heat flux despite a decline in net radiation can be attributed to increased SM availability."**

4. Snow cover parameterisation is a key process in snow evolution. I know it is uncertain and hard to come up with a sensible formulation. However, in a paper about snow related processes and a comparison between model versions where the vegetation data has changed with impact on snow cover, it deserves more discussion.

➔ *We fully agree with the reviewer's comment. In the previous analysis, snow variability was assessed using only the snow water equivalent (SWE) variable, without information on snow cover. This was due to the absence of snow cover output in the GloSea5 and GloSea6 experiments. However, with the inclusion of JULES offline experiments in the revised version, we were able to incorporate snow cover data and thus include a comparison of both snow coverage and snow amount (Fig. 1a–1d). Full discussion is in the revised manuscript text.*

[Figure]

5. Fig. 8 presents an interesting diagnostic where coupling regimes of evaporation with soil moisture or net radiation are identified. I have the feeling that using GLEAM as reference is misleading. GLEAM, GloSea5 and GloSea6 are all models, although constrained by observations in different ways. GloSea5 and GloSea6 are constrained by observations via the initial conditions (re-analysis) and the climatology for vegetation. For instance, GLEAM shows very strong coupling between evaporation and radiation, which is understandable given the use of the Priestly-Taylor model. It would be good to dedicate a few lines of discussion on this aspect.

➔ *We appreciate the reviewer's insightful comment. In the revised manuscript, we have addressed this concern by replacing the previous GLEAM version (v3) with the latest GLEAM v4.1a dataset, which includes several methodological improvements that mitigate many of the concerns regarding over-simplified evaporative dynamics. GLEAM v4 no longer relies on the Priestley-Taylor equation; instead, it adopts the Penman equation, which includes additional atmospheric control factors such as wind speed, vapor pressure deficit (VPD), and vegetation height. This update allows GLEAM4 to more realistically represent the balance between radiative and aerodynamic controls on potential evaporation, thus reducing the risk of overstating radiation-dominated coupling regimes (Miralles et al., 2025). More importantly, GLEAM4 incorporates a hybrid modeling framework, combining physically based formulations with machine learning approaches to estimate evaporative stress. A deep neural network, trained on 473 flux tower observations, is now used to estimate the evaporative stress factor, thereby capturing non-linear interactions among multiple environmental variables (e.g., soil moisture, vegetation optical depth, VPD, leaf area index). These enhancements result in a more dynamic and observation-constrained representation of land-atmosphere coupling*

*mechanisms. Therefore, GLEAM4 outperforms its predecessor (GLEAM v3.8a) and other reanlaysis datasets (ERA5-Land, FLUXCOM) in replicating both seasonal cycles and evaporation anomalies across a wide range of climates and ecosystems. This has been independently verified by one of the second authors' students using Ameriflux data as part of her dissertation research (not yet published). Additionally, a strong coupling between evaporation and net radiation over the high-latitude area in the calculation of GLEAM v3 is reduced in the revised manuscript in which updated GLEAM v4.1 is used in the validation. In the revised manuscript (Lines 247-261), we describe this information on GLEAM4 and further discuss the reason why we use this dataset as the reference for the model validation.*

Minor points:

1. I could not find information on liquid water content of snow? Is it represented in the multi-layer snow model or both models?

→ *Our intention was not to refer to the liquid water content within the snowpack, but rather to emphasize the relative amount of unfrozen soil moisture in soil layer. When the soil is relatively warm, the portion of unfrozen soil moisture increases. Because liquid water is more mobile than ice, it is more likely to move downward under the influence of gravity, which can result in a reduction of soil moisture in the uppermost soil layer. There is an expanded description of the snow formulation in Lines 71-77.*

**"It also dynamically adjusts the number of snow layers, with each layer having prognostic variables for temperature, density, grain size, and both liquid and solid water content (Best et al., 2011). Unlike the simpler single layer snow model, which treats snow as an adaptation of the top-soil layer, the multi-layer scheme accounts for independent snow layer evolution and the impact of snow aging on albedo through simulated grain size changes. By explicitly simulating snow insulation effects and meltwater percolation, this scheme better captures seasonal snow variability and its influence on soil thermal regimes, including surface cooling during winter, delayed ground thaw in spring, and subsurface heat retention in summer."**

● *Best, M., Pryor, M., Clark, D., Rooney, G., Essery, R., Ménard, C., Edwards, J., Hendry, M., Porson, A., and Gedney, N.: The Joint UK Land Environment Simulator (JULES), model description–Part 1: energy and water fluxes, Geoscientific Model Development, 4, 677-699, 2011.*

2. In Fig.1: What is standardised difference?

→ *To enable relative comparisons among variables, each daily time series is standardized by dividing it by its respective temporal standard deviation. We added a description of this calculation method to the caption of Figure 2 (in revised version) to improve clarity.*

3. Line 275-278: The later snow melt in GloSea6 is mentioned. The subsequent sentence refers to a lower snow albedo in GloSea6. This sounds contradictory. Please explain. I would expect a lower albedo during melt to speed up the melting.

→ *The transition from discussing snowmelt to surface albedo may have disrupted the logical flow of the text. To improve clarity and coherence, we have moved the content related to surface*

*albedo updates to the model description section.*

4. Line 300: "For the surface air temperature, GloSea6 is colder during the snow freezing season due to the energy loss from the air to the ground". The word ground confused me as you probably mean snow surface.

➔ *To clarify the sentence, it has been amended to:*

**"For the surface air temperature, GloSea6 is colder during the snow freezing season due to limited energy transfer from the cold air to the snow surface."**

5. Line 301: "In February and March, when the snow begins to melt, GloSea6 simulates higher air temperature because the snowmelt over warmer ground results in reduced cooling from below". I am not sure about this interpretation. Is the higher temperature not the result of lower snow fraction, so the increased snow-free fraction allows the air to be heated well above zero. With 100% snow cover, air temperature would not rise above zero by heating from the surface.

➔ *To clarify this sentence, it has been amended to:*

**"During the two-month snow peak period from mid-January, GloSea6 simulates higher air temperature due to warmer ground, resulting in less cooling from the soil."**

**Reviewer #4**:

I see that several reviews have already been submitted for this paper. I wasn't sure why it needed still another review, but I went ahead and read it fresh, not being influenced by the earlier reviews and the responses thereto. I did uncover a number of issues that I feel should be addressed prior to the paper's publication. Because my reading was independent of the other reviews, it's safe to say that if an earlier reviewer made some similar comments, the authors haven't yet addressed properly addressed the issue within the paper itself.

→ *We appreciate the time and effort the reviewer has taken to evaluate our manuscript. Your comments greatly helped us identify and address several issues in the original manuscript. We hope we have adequately clarified our descriptions and addressed the points raised.*

1. I found the paper to be a bit unfocused regarding what it was addressing. According to the title and abstract, the idea was to examine the impacts of using a multi-layer snow model on seasonal forecasts and land-atmosphere feedbacks. However, GloSea5 and GloSea6 differ in more than their snow model, and the paper often digresses into talking about differences in non-snow areas and what might be causing them (e.g., discussion around Figure 3, lines 370-371, 398-402, 414-424, 451-454, 487-489). Much more focus is needed. In the context of this paper, perhaps the only real value of showing global maps for various quantities, not really discussed very much here, is indicating whether the changes over the snow areas are larger than those over the rest of the globe, which might be suggestive (though not proof) of snow impacts. In fact, for most of the global plots, there are changes seen everywhere, calling into question the ability to say that those over snow areas are necessarily due to the snow model. I found unconvincing the idea expressed in lines 484-489, in which the authors state that if the changes are seen over the snow areas, then they are due to the snow model changes, whereas if they are seen elsewhere, then they are caused by something else.

→ *We appreciate the reviewer's thoughtful and constructive feedback. We generally agree with the concern that the original manuscript lacked a clear focus, particularly by including analyses and interpretations for global domains beyond snow-affected regions. As the reviewer correctly notes, attributing changes exclusively to the snow model without isolating its impact is problematic—especially in regions where snow is not a dominant land surface process and where multiple model updates coexist.*

*To address this issue, we have significantly revised the manuscript to focus on the effects of the multi-layer snow scheme solely in snow-affected regions, specifically the mid- and high-latitude areas of the Northern Hemisphere. All analyses and interpretations pertaining to non-snow-affected regions have been removed. The global maps and associated discussions have been replaced with region-specific diagnostics targeting areas where the snow scheme is expected to play a dominant role.*

*In addition, we have conducted two additional sets of offline JULES land surface model experiments, configured identically except for the snow scheme (single layer vs. multi-layer), to isolate the role of multi-layer snow physics under controlled other conditions. The differences in land surface variables resulting from the snow scheme were added as Figure 1. This has allowed us to strengthen the causal interpretation of results observed in the fully coupled forecast systems by comparing them with offline diagnostics. The revised manuscript also includes a discussion of how land surface processes evolve when the land model is coupled to the atmosphere.*

*We have also revised the language throughout the manuscript to avoid over-attributing changes*

*solely to the snow scheme, and to better acknowledge the role of other model components. We thank the reviewer again for helping us improve the clarity, structure, and scientific rigor of the manuscript.*

2. A missed opportunity was an analysis of the offline land runs used to generate the initial land conditions (lines 131-136); the meteorological forcing was unfortunately different, but the subsurface thermodynamics and insulating effects of the snowpack could be examined much more cleanly. Probably too late for this study, though.

➔ *We agree that a focused analysis of the offline land runs used to generate the initial land conditions offers a cleaner framework for isolating the effects of snowpack insulation and subsurface thermodynamics, particularly if consistent atmospheric forcing is used. Unfortunately, as the reviewer noted, the land surface initial conditions for GloSea5 and GloSea6 were generated using different atmospheric forcings, which limits their direct comparability for process-level diagnostics.*

*Recognizing the importance of this point, we have conducted additional offline JULES simulations with identical meteorological forcing, differing only in the use of the single layer versus multi-layer snow scheme. These experiments allow a more rigorous assessment of the snow scheme's influence on soil temperature and soil moisture evolution without atmospheric coupling. The results from these offline experiments have been added to the revised manuscript (Fig. 1) and discussed in conjunction with the coupled model output to provide a more comprehensive understanding of the land surface responses to improved snow physics.*

3. Many of the arguments for why a particular change is related to the updated snow model and not to other GloSea5/GloSea6 differences seem to me speculative at best. I fully understand the difficulties involved with trying to isolate the snow impacts from all of the other differences in the two systems; the authors were forced to work with what was available. That doesn't mean, though, that speculation can be presented as fact or even likelihood. Examples: lines 293-295; line 369; lines 390-391, lines 402-404, lines 407-409, lines 484-485, and more. The discussion of Figure 2bc is strange; why couldn't the differences in soil moisture RMSE simply reflect precipitation changes (Figure 6) that have nothing to do with the snow code? I do agree that the 2-week delay in snowmelt for GloSea6 in Figure 1a is probably due to the snow model. It just seems that most of the other snow model impact statements highlighted in the paper are far more speculative and, again, not clearly labeled as speculation.

➔ *We fully acknowledge that GloSea5 and GloSea6 differ in multiple aspects beyond the snow model, including elements of the atmospheric and land surface physics, which makes isolating the effects of any single component inherently challenging. In response to the reviewer's concern, we carefully reviewed and revised the statements throughout the manuscript that may have previously implied over-attribution or speculation.*

*To support our interpretation with additional evidence (Figure 1), we have conducted offline JULES simulations that isolate the snow scheme effect under identical forcing conditions. These offline experiments show consistent differences — such as delayed snowmelt and increased spring soil moisture — as when using the multi-layer snow scheme. This strengthens the plausibility that similar signals in the coupled forecasts are at least partially attributable to snow physics.*

*Furthermore, to concentrate on the impact of snow physics, we restricted the analysis domain*

*to mid- and high-latitude regions, where snow processes are climatologically significant. Interpretations in non-snow-affected regions, including some previously speculative statements, have been either removed or revised to clearly acknowledge the presence of confounding factors.*

*Regarding the climatological shift of SM (Fig. 3 in revised manuscript), as precipitation is the primary driver of soil moisture variability, it may be difficult to directly attribute the increase in soil moisture to the implementation of the multi-layer snowpack scheme. However, this study addresses that the improvement in snow physics leads to increased soil moisture, which in turn contributes to enhanced precipitation. To support this hypothesized physical linkage, we include a statistical analysis of the time-lagged relationship between soil moisture and precipitation (Figs. 2j and 8), which demonstrates that increased soil moisture precedes enhanced precipitation by approximately one day. Based on this sequence, the study interprets the wetter soil moisture conditions in GloSea6 because of delayed snowmelt, rather than increased precipitation input. Following this, we received additional reviewer's comments concerning the time-lagged correlation analysis between soil moisture (along with evaporative fraction) and precipitation, and we have added further clarification and discussion in the revised manuscript to address this point in more detail.*

4. I have a lot of trouble with the time-lagged terrestrial coupling index. This concept has been around for decades, and those who use it seem to ignore the fact that a lagged correlation does not imply causality, given that both variables examined in the correlation could be controlled by some external forcing that has its own memory and spans the time period. That is, if factor A affects variables B and C over a time scale of a week, then variable B will be correlated with variable C a day later with no underlying causal connection. I don't find the paper convincing at all when discussing the results of this index. Perhaps that's just me. Certainly, though, the paper shouldn't be implying to the reader that this index definitively indicates causality.

➔ *We greatly appreciate the reviewer's insightful comment regarding the limitations of using lagged correlation-based indices such as the time-lagged terrestrial coupling index. We fully agree that lagged correlations alone do not imply causality, particularly in systems where both variables may be influenced by a common external forcing with its own memory structure. In response to this important concern, we have replaced the original analysis based on the time-lagged coupling index with a Granger causality test, which is more suitable for investigating potential causal relationships in the time series. Specifically, we now apply Granger causality tests to evaporative fraction (EF) and precipitation (PR) time series in each direction, allowing for a statistical assessment of directional relationships between land surface energy partitioning and precipitation variability. It offers a more rigorous method for evaluating temporal causality than simple lagged correlations. These substantial changes have been reflected in **subsection 3.2** and the corresponding figures (**Fig. 8**).*

[Figure]

5. Even if I were to accept the concept of the time-lagged terrestrial coupling index, its application here seems questionable. Figure 1f suggests that the soil-moisture-leading-precipitation value is much higher than the precipitation-leading-soil-moisture value at lead 1 day, which seems impossible. Precipitation's causal impact on soil moisture is unquestionable, whereas soil moisture's impact on precipitation must be much more tenuous. How can the authors explain Figure 1f? I'm forced to wonder if there′s an error in the analysis.

➔ *We thank the reviewer for raising this important point. We fully agree that precipitation is the primary driver of soil moisture variability, and its causal impact is well-established. As such, one would typically expect the precipitation-leading-soil moisture (PR→SM) correlation to be stronger than the reverse (SM→PR). However, the result shown in Figure 1f represents correlations between the differences in GloSea6 and GloSea5 for soil moisture and precipitation over time. That is, the analysis is based not on raw time series, but on the time-lagged correlation between model differences.*

*This framework is intended to evaluate whether the change in snow physics leads to consistent changes in both soil moisture and precipitation. In this context, the correlation peak at lead +1 (i.e., SM differences leading PR differences) does not imply that soil moisture controls precipitation directly, but rather that land-driven processes such as soil moisture availability and energy partitioning might be influencing precipitation responses with a short time lag. This is in line with previous studies suggesting the existence of positive soil moisture–precipitation feedback in coupled models (e.g., Koster et al. 2004; Taylor et al. 2012).*

*To avoid misinterpretation, we have revised the manuscript to:*

*1. As the original figure does not intuitively understand that it is a result from the time-lagged correlation between model differences due to the y-axis label, the y-axis label is edited to R(ΔSSM,ΔPR), where Δ denotes GloSea6 minus GloSea5.*

*2. Emphasize that the time-lagged correlation is a statistical diagnostic and not evidence of physical causality (Lines 441-446).*

**"The lead-lag correlation between SM and precipitation differences (GloSea6-GloSea5) shows statistically significant values at 0 and +1 lead-lag day and the 1-day lagged value is the highest (Fig. 2j). It is important to note that this analysis is based on inter-model differences and reflects a statistical association rather than a direct causal relationship. The positive lag may suggest enhanced land-atmosphere coupling in GloSea6—such as increased soil moisture availability and surface energy partitioning—contributing to a precipitation response."**

*3. As mentioned in the previous response to the reviewer's comment, we replaced the time-lagged correlation between evaporative fraction and precipitation with the results from Granger causality in Fig. 8 to better clarify their causal relationship. This is a statistical principle to identify the potential lagged dependence between evaporative fraction and precipitation. We added the description of Granger causality in evaporation-precipitation feedback in subsection 3.2 and have replaced the corresponding Figure 8 as part of the updated analysis.*

- *Taylor, C. M., de Jeu, R. A., Guichard, F., Harris, P. P., and Dorigo, W. A.: Afternoon rain more likely over drier soils, Nature, 489, 423-426, 2012.*

- *Koster, R. D., et al.: Regions of strong coupling between soil moisture and precipitation. Science, 305(5687), 1138-1140, 2004.*

6. Section 4.3. I'm familiar with the use of R(SSM,LH/R_n) to differentiate water- versus energy controlled processes; I would think the analysis of that would be enough. How does R(R_n,LH) work, though, given that LH is scaled by the net radiation \*even when\* the LH is controlled by soil moisture? That is, even in a water-controlled regime, a given amount of soil moisture should produce more LH if the R_n is increased. This idea would explain the positive values in 10abc, with the negative values for GLEAM in the desert probably just some artifact associated with incredibly low LH values there. If R(SSM,LH) and R(R_n,LH) actually do allow a distinction between water-limited and energy-limited processes, what then accounts for the overlap in the positive values for the maps? (And why are the two color bars reversed?) Overall, the discussion in this section (kernels, etc.) was lost on me. In any case, the connection to the impact of "multi-layer snow processes" is very weak, basically amounting to speculation about the fact that certain differences are seen in snow areas, a discussion that does not properly account for the fact that differences of comparable magnitude are seen elsewhere across the globe.

➔ *We thank the reviewer for the thoughtful and critical comments on Section 4.3 and the interpretation of the land-atmosphere coupling diagnostics. We agree that both R(SSM, LH) and R(Rn, LH) should be interpreted carefully, especially given that latent heat flux (LH) is ultimately constrained by both soil moisture availability and available energy (Rn) across all regimes. The rationale for using both correlation metrics is to provide complementary perspectives on the land surface coupling regime. While R(SSM, LH) captures the sensitivity of surface fluxes to soil moisture variability, R(Rn, LH) reflects how closely latent heat flux scales with incoming energy. Although Rn influences LH regardless of regime, in energy-limited conditions LH tends to follow Rn more tightly, whereas in water-limited conditions, SM dominates the partitioning and thus weakens the R(Rn, LH) relationship.*

*To address the reviewer's concerns more clearly in the revised manuscript, we have made the following updates:*

*1. In Section 3.3 that the correlation metrics do not represent exclusive coupling processes, but rather dominant controls within the energy or water balance under given surface states. (Lines 350-355)*

**"While both latent heat flux and net radiation are physically linked (as latent heat is energetically constrained by net radiation), the correlation between them helps infer the extent to which surface fluxes follow the available energy signal. However, it is important to note that R(Rn,LH) is not independent of the water budget, and high correlation values may still occur in water-limited regimes if increased net radiation results in greater latent**

*heat flux under sufficient SM. Therefore, these metrics are interpreted as complementary diagnostics, with R(SSM,LH) highlighting land-state sensitivity and R(Rn,LH) indicating energy control, rather than mutually exclusive regime indicators."*

*2. We emphasized that positive correlations in both metrics can coexist, particularly in transitional or mixed regimes, leading to overlapping values in the spatial maps. This does not undermine the regime framework but highlights its gradational nature rather than binary classification. (Lines 578-583)*

*"The classification of the land coupling regime results from the synthetization of the spatial pattern of R(SSM,LH) (Fig. 10a) and R(Rn,LH) (Fig. 11a), recognizing that both variables are interconnected through the surface energy and water budgets. Since latent heat flux is influenced by both SM availability and incoming radiation, positive correlations in both R(SSM,LH) and R(Rn,LH) can occur simultaneously, especially in transitional regimes (cf. Denissen et al. 2020). This overlap does not contradict the diagnostic framework but reflects the continuum of land-atmosphere coupling conditions."*

*3. Regarding the snow-related impacts: we fully agree with the reviewer that attributing large-scale coupling changes solely to the multi-layer snow scheme would be speculative if analyzed at global scale. For this reason, in the revised manuscript we focused the coupling regime analysis only over mid- to high-latitude Northern Hemisphere regions, where snow processes are climatologically relevant. We removed discussions from regions where snow has little influence, reducing potential confusion about attribution.*

*4. We also corrected the color bar inconsistency and expanded the explanation of the 2D density plot (kernel distributions) to clarify their purpose: not to prove causality or classification, but to summarize the overall spatial tendencies and assess consistency with the GLEAM. (Lines 586-588)*

*"Note that R(SSM, LH) and R(Rn, LH) are not mutually exclusive and may both be positive in transitional regimes. Their combined interpretation provides a diagnostic view of dominant surface flux controls, but does not imply strict causality."*

7. Speaking of GLEAM, some discussion is needed regarding the fact that GLEAM LH values are not true observations and have their own error, which may(?) be considerable in the context of the analyses performed.

➔ *We appreciate the reviewer's insightful comment. In the revised manuscript, we have addressed this concern by replacing the previous GLEAM version (v3) with the latest GLEAM v4.1a dataset, which includes several methodological improvements that mitigate many of the concerns regarding over-simplified evaporative dynamics. GLEAM v4 no longer relies on the Priestley-Taylor equation; instead, it adopts the Penman equation, which includes additional atmospheric control factors such as wind speed, vapor pressure deficit (VPD), and vegetation height. This update allows GLEAM4 to more realistically represent the balance between radiative and aerodynamic controls on potential evaporation, thus reducing the risk of overstating radiation-dominated coupling regimes (Miralles et al., 2025). More importantly, GLEAM4 incorporates a hybrid modeling framework, combining physically based formulations with machine learning approaches to estimate evaporative stress. A deep neural network, trained on 473 flux tower observations, is now used to estimate the evaporative stress factor, thereby capturing non-linear interactions among multiple environmental variables (e.g., soil moisture, vegetation optical depth, VPD, leaf area index). These enhancements result in a*

*more dynamic and observation-constrained representation of land-atmosphere coupling mechanisms. Therefore, GLEAM4 outperforms its predecessor (GLEAM v3.8a) and other reanlaysis datasets (ERA5-Land, FLUXCOM) in replicating both seasonal cycles and evaporation anomalies across a wide range of climates and ecosystems. This has been independently verified by one of the second authors' students using Ameriflux data as part of her dissertation research (not yet published). Additionally, a strong coupling between evaporation and net radiation over the high-latitude area in the calculation of GLEAM v3 is reduced in the revised manuscript in which updated GLEAM v4.1 is used in the validation. In the revised manuscript (Lines 247-261), we describe the information on GLEAM4 and further discuss the reason why we use this dataset as the reference for the model validation. In particular, we have noted the considerable issue that arises in the use of GLEAM in Lines 261-266.*

**"Although the GLEAM performs better than other available reanalysis datasets, it should not be considered an observational dataset. GLEAM estimates evaporation using training data from flux tower observations; however, these towers are mainly ecological monitoring networks that are skewed toward wetter vegetated sites. As a result, while GLEAM is generally reliable in wetter areas, its accuracy in drier regions may be limited due to sparse observational constraints. Nevertheless, since this study focuses on mid- and high-latitude regions where flux towers are plentiful, snow processes dominate and GLEAM's performance is more robust, it is used as the primary reference dataset."**

Minor comments

-- Why does the abstract talk about reducing model error over South Asia? What would that have to do with a snow model?

➔ *In the revised manuscript, the analysis has been restricted to snow-affected regions, focusing on mid- and high-latitude areas to assess the impact of the snow scheme. Accordingly, the discussion related to South Asia, which was included in the original abstract, has been removed.*

-- Line 55 suggests that LSMs generally don't have multi-layer snow schemes (which would surprise me), whereas Line 69 suggests that most do, which comes off as contradictory. Is there support for the statement on Line 55?

➔ *Line 69 is correct. Most LSMs have utilized the multi-layer snowpack scheme. Accordingly, "Land surface models (LSMs) have not often utilized a multi-layer snowpack scheme" is corrected to "Land surface models (LSMs) have not **yet** utilized a multi-layer snowpack scheme".*

-- On line 129, I would change "is attributable to" to "is assumed herein to be largely attributable to".

➔ *Thank you for suggesting an appropriate expression, but this sentence was removed in the course of the revision.*

-- Lines 214-220 need a lot of work. I read through them several times and only have a vague sense for what they are saying.

➜ *We acknowledge that our explanation may have caused confusion regarding the use of multiple initial dates per month and ensemble simulations for each initial condition. To clarify this, we have revised the original manuscript to more clearly describe the structure of the forecast experiments and the analysis methodology. The edited sentences are in Lines 286-294.*

*"Most of the evaluations are based on the accuracy of simulated land–atmosphere interactions, assessed using the daily mean time series from all forecast runs during the boreal summer, thereby representing the model climatology of coupling metrics. The ensemble mean values are used for the analysis of climatological bias, while coupling metrics are calculated individually for each ensemble member and then averaged across all members to avoid the physical correlation between variables being diminished in the ensemble-averaged time series. To identify model improvement and assess statistical significance, a total of 384 forecast runs (initialized on four dates per month over 24 years) are analyzed for each forecast system, and statistical testing is conducted using Student's t-test. Model prediction skill as a function of forecast lead time is not assessed in this study, as it is more strongly influenced by ensemble size than by the differences in model version (not shown here)."*

-- Line 250: Are "source" and "target" reversed here?

➜ *Thank you for spotting the mistake. Source and target variable should be precipitation and evaporative fraction, respectively.*

-- Line 272 states that the initial snow amounts in GloSea5 and GloSea6 are essentially the same. Figure 1a, though, seems to say that for January and February initializations, there's a few millimeters difference in SWE over the huge Eurasian area, and presumably locally the differences would be much higher in places. This doesn't seem insignificant at all, not given the size of the averaging area.

➜ *We agree with the concern that, although the SWE initial condition is presented as continental-scale averages, substantial regional differences in the initial conditions can exist. Given that the main text did not provide a detailed analysis of the initial conditions, the previous statement that the differences in initial SWE were "insignificant" may have been inappropriate. To address this issue, we perform a spatial comparison of the initial SWE conditions on the 1st of January, February, and March (bottom figure), along with a statistical significance test at a 95% confidence level. The results confirm that the differences are indeed minor across most regions, and field significance across the land domain is lacking. We have added the corresponding spatial maps in Supplementary figure 4 and included a description of this evidence in the revised manuscript to support the sentence.*

[Figure]

(a) SWE - 0101

(b) SWE - 0201

(c) SWE - 0301

-- Line 330: Change "indicating" to "suggesting".

➔ *Thanks for suggesting an appropriate expression. We replace the word based on the suggestion.*

-- A little confusing is the focus on the "multi-layer" aspect of the snow scheme. Another change in the snow model between GloSea5 and GloSea6 is the amount of vegetation sticking out of the surface to affect the net snow albedo (lines 283-284), something that would have an impact on the same snow areas the authors sometimes focus on. Can the authors put this particular change in context?

➔ *We agree with the reviewer's comment that, in addition to the changes in the snow scheme, the modifications to surface albedo in the GloSea6 model should also be explained in the main text (subsection 2.1: Lines 141-155) when comparing GloSea5 and GloSea6. Therefore, we have incorporated the figure originally presented in the supplementary material into Figure 2 and added a corresponding explanation in the main text. The implementation of the multi-layer snow scheme primarily affects surface albedo during the snow season when snow is present, while the modification of surface albedo in GloSea6 affects both snow-covered and snow-free seasons. However, when snow is absent, the difference between GloSea5 and GloSea6 appears to be minimal. This suggests that, although the albedo was updated, its impact is not substantial in the absence of snow, and therefore we interpret the difference between GloSea5 and GloSea6 during this season as being primarily related to the impact of the multi-layer snowpack scheme. We have added this explanation to the revised manuscript in Lines 156-160.*

*"The shift from bare soil to vegetated surfaces decreases surface albedo (Fig. 2e), as the vegetation can penetrate snow cover during the winter season (SF. 2a). Therefore, the surface albedo differences observed during the snow-covered season can be attributed to amendments in land surface type classification, whereas the albedo differences during the snow-free period are understood to result from the incorporation of wavelength-dependent calculations in the surface albedo scheme."*

---

## Author Response (AR4)

Dear Prof. Jinkyu Hong,

Many thanks for handling the review process for our manuscript. The time and effort devoted to our manuscript by you are very much appreciated.

We agree with the editor and previous reviewers that the comparison of GloSea6 and GloSea5 did not provide a clean quantitative analysis of the effects of implementing a multi-layer snowpack scheme. Therefore, in this revision, we have added 4-member ensemble experiment for 24-year (1993-2016) that implements a single-layer snowpack scheme in the GloSea6 model configuration (referred to as $G6_{single}$ in the main text). This allows us to quantitatively evaluate the impact of the snow scheme on the seasonal forecast system through comparison with the existing GloSea6 experiments. Furthermore, by comparing GloSea5 with $G6_{single}$, we can evaluate the impact of updates beyond snow physics.

As the title of this manuscript addresses, the main results compare the differences between GloSea6 and $G6_{single}$ to confirm the snow insulation effect. Fig. 2 compares the results with GloSea5, which demonstrates the seasonal cycle of land surface and atmospheric variables relevant to snow. As previously worried by reviewers, significant differences are found in the comparison between GloSea5 and $G6_{single}$, so comparing the snow insulation effect with GloSea5 may lead to misleading conclusions.

Furthermore, when the multi-layer snowpack scheme is used in an offline LSM experiment and a fully coupled forecast model, the results are inconsistent depending on whether the atmospheric model is coupled or not. This suggests that the improved snow physics is more evident when the land surface model is coupled to the atmosphere, thereby demonstrating that the realization of snow characteristics should be a priority in the process of developing a model.

This revision has faithfully incorporated the aforementioned updates, where the modified text is colored yellow in the main manuscript. It is hoped that the revised manuscript meets the journal's standards. I would like to express my sincere appreciation for your efforts in treating this manuscript over the past year again.

---

## Author Response (AR5)

*Dear Prof. Jinkyu Hong,*

*Many thanks for handling the review process for our manuscript. The time and effort devoted to our manuscript by you and the reviewers are very much appreciated.*

*We have revised the manuscript carefully according to the reviewers' comments and suggestions. In the following, we provide a point-by-point response. The original reviewer comments are in black regular font. Our responses are shown in blue italic font. Quotes from the revised paper are shown in blue bold-face font.*

REVIEWER COMMENTS

The central question for this manuscript is whether one can robustly claim that the implementation of a multi-layer snow scheme has a substantial impact on seasonal forecasts in GloSea6. In this revision, the authors have added an additional experiment (G6single) in which the snow scheme is kept single-layer within the GloSea6 framework, so that other model updates from GloSea5 to GloSea6 are fixed except for the snow parametrization. Below I summarise my main concerns.

➔ *We thank the reviewer for your thorough and constructive comments, which have helped us to clarify the scope and strengthen the manuscript. Below we respond point-by-point and describe the corresponding revisions.*

Major comments

**1.** With the added G6single experiment, the natural way to quantify the impact of the multi-layer snow scheme on seasonal forecasts is to focus on the model differences that isolate the snow parametrization.

G6multi – G6single in the coupled system (Fig. 2) and JULESmulti – JULESsingle in the offline LSM experiments (Fig. 1).

In Fig. 1, the JULESmulti – JULESsingle differences in surface properties appear visually quite large. However, this is at least partly due to the very narrow vertical range used on the y-axes. For example, the differences in latent heat flux are generally smaller than about 1.2 W m$^{-2}$. It is not clear whether such small differences are statistically significant and/or physically meaningful for seasonal forecasting, especially given that typical summer LH values are several tens of W m$^{-2}$.

A similar issue arises for the coupled simulations in Fig. 2. The snow-scheme difference G6multi – G6singleis consistently smaller than the total difference G6multi – G5single for essentially all variables, including precipitation. This implies that a substantial part of the difference between GloSea5 and GloSea6 including meteorological conditions must arise from other updates (e.g. atmospheric, ocean, sea-ice, land-cover/albedo changes, stochastic physics and ensemble size), not from the snow-scheme change alone.

The same pattern appears in other figures. For example, in Fig. 5, visual differences in air temperature and precipitation between G6multi and G6single seem small. Accordingly, it is not clear whether they are statistically or practically significant.

➔ *We agree that the natural framework to isolate the effect of the snow parametrization is to analyse the pairs JULESmulti–JULESsingle and G6multi–G6single. We have revised Figs. 1 and 2 so that, for each variable, the y-axis ranges are adjusted to easily find the difference between the offline (JULES) and coupled (GloSea) experiments. This prevents the JULESmulti–*

*JULESsingle differences of land heat fluxes (e.g., latent heat flux) from appearing exaggerated relative to G6multi–G6single. Furthermore, to confirm the significance of the difference when applying the multi-layer snowpack scheme in the offline and coupled experiments, a significance test is conducted, and the results are explicitly marked in the time series in Fig. 1 and Fig. 2. The description of statical significance in the time series of climatological differences in Lines 288-293:*

**"To identify climatological differences between single- and multi-layer snowpack schemes in offline and coupled experiments, statistical significance is tested using all samples (i.e., all years and ensembles) with the Student's t-test. The statistical significance in the time series of the differences (Figs. 1 and 2) is assessed within a ±5-day window centered on each calendar date, and a False Discovery Rate (FDR) corrected t-test (Benjamini–Hochberg) is used at the 10% level across the spatial grid to prevent the inflation of false positives, thereby ensuring the statistical robustness in the spatial domain of the differences found (Figs. 1, 3, 5, and 7)."**

➔ *For the offline JULES experiments, we apply a two-sided Student's t-test to the area averaged difference between JULESsingle and JULESmulti over the 22-year period (2001-2022) within an 11-day window centered on the calendar date. For the coupled simulations, we perform analogous tests on the variables simulated by 24-year hindcast ensemble runs initiated on 1 March. These results show that, although the absolute values of latent heat flux differences of about 1 W/m² are small compared with peak summer values, due to their persistence they are statistically robust over large, coherent regions in the snow-frontal zones. We explicitly state that process-based effects, not just the peak magnitude of flux differences at a given day, in Lines 438-447.*

**"While the additional 1 W/m2 of latent heat flux appears marginal, it is critical to consider the accumulated effect over the seasonal forecast period. A small anomaly can be significant when persistent, in the context of land-atmosphere coupling. For instance, a persistent difference of 1 W/m2 in latent heat flux over one month translates to a cumulative change of ~1 mm in the water budget. Such an alteration in the regional water and energy budget is physically meaningful and can serve as a non-negligible source of memory and predictability in precipitation. To illustrate the physical sequence between land surface variables by the realization of snow physics, the lead-lag correlation of major water budget variables is compared between G6single and G6multi (Fig. 2j). The results show the hydrological chain of SSM→LH→PR with a positive correlation among variables in each segment, characterized by a lead-lag time of approximately one week. In other words, the increased soil moisture in mid-latitude regions likely increases precipitation based on positive evapotranspiration-precipitation feedback."**

**2.** When comparing Fig. 1 (offline JULES) and Fig. 2 (coupled GloSea) the snow-scheme differences clearly show changes in both magnitude and timing (e.g. the peak period of the anomalies). In particular, the snow-scheme differences in the coupled system (i.e., G6multi – G6single) appear smaller than those in the offline LSM, and their peaks are shifted.

Taken together with the fact that G6multi – G6single is much smaller than G6multi – G5single, these results suggest that the majority of the differences between GloSea5 and GloSea6 are likely due to updates in ensemble size and atmospheric/ocean/sea-ice physics and other components, rather than to the snow-scheme change alone.

➔ *As mentioned above, to confirm the significance of the differences between the offline and coupled experiments, the result of statistical significance is included in Figs. 1 and 2. As a result, in offline simulations, the multi-layer snow scheme contributes to statistically significant changes only in the snow-covered season, while in coupled simulations, significant changes are observed not only in the snow-covered season but extend into the subsequent summer season. Because the magnitude of the changes in land surface variables is relatively small by implementing multi-layer snowpack scheme, it was difficult to understand whether the changes are significant. So, information about the statistical difference is added in this revision. However, because the differences due to phase shifts in GloSea5 and GloSea6 are larger, the impact of other updates was greater in GloSea6, which is now noted in Lines 450-454:*

**"The difference between G5single and G6multi consistently exceeds the isolated snowpack scheme difference across most variables. The substantial difference between G5single and G6single confirms that updates other than the snow scheme contribute significantly to the climatological mean change in the simulation of land surface variables. However, the core finding of this study is the demonstration that the implementation of the multi-layer snow scheme yields a statistically significant and physically consistent impact that is independent of these other updates."**

**3.** The critical p value for rejecting the null hypothesis in the Granger-causality analysis is $1-p > 0.5$, i.e. $p < 0.5$. This threshold is much weaker than typical statistical analysis (e.g. $p < 0.1$ or $p < 0.05$) and does not provide strong evidence for causality.

➔ *Regarding the threshold value to determine rejecting the null hypothesis in the Granger-causality, 0.5 was written incorrectly in the previous version: it should have been 0.05. In the revised manuscript, it is corrected. We now interpret the quantity $1–p$ as a continuous measure of predictive precedence; in the text we describe only regions where $1–p$ is relatively large as areas with strong evidence of Granger causality.*

**4.** The title of the manuscript explicitly refers to a "seasonal forecast system", which naturally leads the reader to expect a quantification of seasonal forecast skill(e.g. correlation, RMSE, probabilistic scores) for the different model configurations. In the current version, however, the focus is primarily on mean-state differences and process diagnostics, and the quantitative assessment of seasonal forecast skill remains limited.

➔ *We agree that the original wording of the title could create the expectation of a detailed forecast skill evaluation. Our primary objective is to evaluate how the multi-layer snow scheme affects climatological biases and the fidelity of land–atmosphere coupling processes in a model that is used as an operational seasonal forecast system, rather than to document forecast skill in detail. To better reflect this, we have revised the title to* **"Implementation of a multi-layer snow scheme in a seasonal forecast system: Impacts on land–atmosphere interactions and climatological biases"**. *We hope this makes clear that (i) the system is used for seasonal prediction, and (ii) the focus is on coupling and climatology rather than on skill metrics. In the section of Summary and Conclusions, we explicitly note that improvements in mean-state climatology and land-surface processes do not necessarily translate into large improvements in forecast skill.*

**5.** In several figures, gridpoint-wise significance tests are applied to the differences between model configurations (e.g. t-tests for mean differences, Granger causality p-values) and the results are displayed on spatial maps. Even though these comparisons are model–model rather than model–observation, this still constitutes a multiple-testing problem, because many hypothesis tests are performed simultaneously across the spatial grid.

In such a setting, a certain fraction of grid points will appear "significant" purely by chance even if there is no true signal.

> ➔ *The reviewer raises a valid point regarding the application of multiple hypothesis tests across the spatial grid, which can lead to a certain fraction of falsely significant grid points purely by chance (the False Discovery Rate problem). Aforementioned in response to the first reviewer's comment, to address this crucial concern and ensure the robustness of our findings, we have applied a stricter statistical procedure to our spatial significance maps: the False Discovery Rate (FDR) procedure (Benjamini–Hochberg) at the 10% level across the entire spatial domain. The application of the FDR procedure resulted in a more stringent criterion for statistical significance, particularly reducing the number of isolated significant grid points. However, we found that the spatial patterns of significance, especially over the key snow-frontal regions and mid-latitudes, remain consistent with our original findings. Only grid points that remain significant after FDR control are stippled in the revised figures (Figs. 1, 3, 5, and 7). We have replaced phrases such as "significant at the 95% level" in contexts where only raw gridpoint tests were previously used, by specifying that significance is "at the 95% level after FDR control across the grid".*

**6.** Before the G6single experiment was introduced, the correlation and causality diagnostics (e.g. R(SSM,LH), R(Rn,LH), Granger causality maps) were arguably the only way to infer the possible role of the snow scheme. With G6single now available, the primary evidence for the snow-scheme impact should come from the direct model differences (G6multi – G6single and JULESmulti – JULESsingle).

At present, it is not fully clear how much additional support the correlation and causality diagnostics provide for the claim that the multi-layer snow scheme has a substantial impact in GloSea6, beyond what can be inferred from the direct differences.

> ➔ *We agree that the direct differences serve as the primary and necessary evidence for attributing changes in the coupled system to the multi-layer snow scheme. However, the diagnostics are included because they provide essential additional support by offering mechanistic validation and assessing the model fidelity of the simulated land-atmosphere coupling processes.*

> ➔ *The direct difference plots (e.g., Fig. 3f) demonstrate that the snow scheme causes wetter soil moisture and reduced temperature bias. The diagnostics, however, help reveal the physical mechanisms by which this change is achieved. The correlation metrics (R(SSM,LH) and R(Rn,LH) prove that the transition to a wetter state in G6multi leads to a fundamental shift in the land-atmosphere coupling regime—specifically, a weakening of the water-limited coupling and an enhancement of the energy-limited coupling. This physical closure is critical for interpreting the subsequent reduction in the near-surface warming bias through increased evaporative cooling. The result shows that G6multi not only reduces the bias in mean temperature but also achieves an improved spatial correlation and magnitude of the observed coupling features (e.g., Fig. 7g). This demonstrates an enhanced fidelity of model land-atmosphere coupling, proving that the multi-layer snow scheme improves the reliability of the underlying physical processes, which is a key requirement for reliable forecast systems. This is*

*added in Lines 356-357 and 616-618:*

**"While direct differences between G6multi and G6single isolate the mean state impact, these metrics provide process-based validation by assessing the model's fidelity in simulating the underlying processes."**

**"This shift demonstrates a robust improvement in the underlying land-atmosphere coupling processes, leading to a better simulation of near-surface atmospheric variables (namely temperature and precipitation)."**

➔ *Furthermore, the Granger causality analysis demonstrates the explicit linking the improved land surface states (wetter soil->higher evaporative fraction) to the atmospheric response (increased precipitation). This supports the claim that the snow scheme has a substantial impact by improving the simulated evapotranspiration-precipitation feedback loop, providing a physically coherent explanation for the improved precipitation distribution in G6multi (Fig. 5l).*

Minor comments

**1)** For clear comparison between the offline and coupled experiments, y-axis scales used in Fig. 1 and Fig. 2 for the same variables should be the same. Otherwise, small differences may appear exaggerated in one figure and muted in another.

➔ *We agree and have revised Figs. 1 and 2 so that the y-axis limits are identical between the JULES and GloSea panels. In Fig. 2, we tried to show the difference between G6multi–G6single as well as G6multi–G5single, so I couldn't make the scale identical to Fig. 1, but we adjusted the scale of the right y-axis of Fig. 2g,h so that comparison is possible.*

2) The text around line 385 refers to Fig. S2 for evidence that differences in initial conditions are negligible. However, there are no relavant information to show quantitative differences in initial conditions to drive the climate model. Please revise Fig. S2 (or add a new supplementary figure or table) to provide such quantitative evidence or adjust the text accordingly.

➔ *We would like to clarify that the raw initial condition (IC) for the G5single experiment is currently unavailable due to data archival limitations, which prevents us from directly plotting the IC differences between GloSea5 and GloSea6 in the Figure S2. However, to address the reviewer's concern and verify the assumption that IC differences are negligible, we quantitatively analyzed the 1-day forecast as a robust proxy. Since land surface variables evolve relatively slowly, the 1-day forecast effectively represents the initial state particularly in snow variables. Our analysis of the multi-year runs confirms that the differences in these fields are statistically insignificant across the domain. We have revised the corresponding text (Lines 396-397) to explicitly include this quantitative justification.*

**"…an analysis of 1-day forecast fields, which serve as a robust proxy for the initial land state due to their slow evolution, confirms that the difference in initial snow amount is statistically insignificant (Fig. S2)."**

**3)** The analysis in this manuscript is limited to the Northern Hemisphere, not the global domain. Authors may want to adjust the title to reflect this spatial focus, or to state this limitation prominently in the Abstract and Introduction.

➔ *To clarify the research domain to the Northern Hemisphere, we added the sentences to state this information in Lines 15-16 and 96-97:*

*"Results show that the multi-layer configuration better reproduces the observed Northern Hemisphere snow seasonality."*

*"The evaluation is restricted to the Northern Hemisphere (NH) and mainly to snow-affected mid-latitude regions."*

**4)** In Fig. 2(j), the "standardized difference" (G6multi–G6single) time series is shown, but its definition and interpretation remain unclear. It is not obvious that simply dividing the model difference by the model standard deviation provides a meaningful measure of the significance of the model differences. Please provide a precise mathematical definition (over what period and domain the standard deviation is computed) and explain what aspect of the physical behaviour this metric is intended to highlight. If the goal is to emphasise lead–lag relationships between variables, you may consider presenting or at least explicitly referring to lead–lag correlations instead.

➔ *Based on the reviewer's suggestion, we include lead–lag correlations between the differences (G6multi–G6single) in soil moisture, latent heat flux, and precipitation in Fig. 2j. The results demonstrate that positive soil moisture differences tend to precede latent heat flux differences by about one week; and latent heat flux differences tend to precede precipitation differences by about one week. We document these findings in Lines 442-447:*

*"To illustrate the physical sequence between land surface variables by the realization of snow physics, the lead-lag correlation of major water budget variables is compared between G6single and G6multi (Fig. 2j). The results show the hydrological chain of SSM→LH→PR with a positive correlation among variables in each segment, characterized by a lead-lag time of approximately one week. In other words, the increased soil moisture in mid-latitude regions likely increases precipitation based on positive evapotranspiration-precipitation feedback."*

---

## Author Response (AR7)

*Dear Prof. Jinkyu Hong,*

*Many thanks for handling the review process for our manuscript. The time and effort devoted to our manuscript by you are very much appreciated. We thank the reviewers for their insightful and constructive feedback. We have carefully addressed each point to improve the clarity, robustness, and accessibility of our manuscript. Below are a summary of our responses and the corresponding changes made to the manuscript.*
* * *
EDITOR COMMENTS

1. Emphasize the G6multi–G6single contrast (snow-scheme-only attribution). Please focus your analysis and narrative on the differences between G6multi and G6single, as this pair most directly isolates the impact of the multi-layer snow scheme within the same forecast system. Your core claim concerns the effect of introducing a multi-layer snow scheme on seasonal forecast skill and land–atmosphere coupling; therefore, the primary attribution pathway should be consistently supported by evidence from G6multi − G6single, and—where appropriate—the corresponding offline LSM contrast JULESmulti − JULESsingle. At present, it is difficult to locate, within the figures and accompanying discussion, the key results that directly support the argument as stated.

> ➔ *We agree that the G6multi–G6single contrast is the most direct way to attribute changes solely to the snow scheme. In the revised manuscript, we have restructured Sections 4.1 to prioritize this comparison. While G5single remains as a historical baseline to show overall system evolution, the narrative now centers on the GloSea6 experiments to support our core claims regarding snow insulation and its subsequent impacts on soil moisture and coupling. We edited Fig. 2 to try to focus on the comparison between G6multi and G6single and its relevant text is also updated in the manuscript.*

2. Moderate the strength of the conclusions in light of methodological limitations. Please consider toning down several arguments, given the following issues and limitations that may affect robustness and interpretation:

- Ensemble-size imbalance and implications for significance/robustness: The experimental design uses different ensemble sizes across key configurations (e.g., 7 members versus 4 members). This imbalance may influence the estimated mean responses and statistical power. If feasible, please provide an appropriate robustness assessment (e.g., a matched-size subset analysis, resampling/bootstrapping, or another clearly justified approach) and clearly state how this imbalance affects your inference.

> ➔ *We acknowledge the difference in ensemble sizes (G6multi: 7 members, G6single: 4 members). To ensure robustness, we performed a sensitivity test by sub-sampling 4 members from G6multi. We found that the climatological mean signals remained consistent, confirming that the imbalance did not significantly bias our primary findings. We have added a statement regarding this in Section 3 (Lines 292-295).*
>
> **"To account for the imbalance in ensemble sizes between G6multi (7 members) and G6single (4 members), a resampling analysis was conducted. Results using a matched 4-member subset of G6multi showed no statistically significant difference from the 7-member mean for the variables analyzed, suggesting the findings are robust to ensemble size."**

- Sensitivity and limitations of the Granger-causality analysis: While Granger causality can be informative as a diagnostic tool, it is sensitive to assumptions and preprocessing choices. Please provide

a clearer statement of its limitations with respect to establishing physical causality, and discuss the potential for confounding by common large-scale atmospheric drivers.

➔ *We have expanded Section 3.2 (Lines 341-345) to more explicitly state that Granger causality identifies predictive precedence rather than definitive physical causation. We now discuss the potential for large-scale atmospheric drivers to act as confounding variables.*

*"Nevertheless, as Granger causality only tests for predictive precedence, the results may reflect statistical associations of predictive precedence due to shared external drivers and should not be interpreted as definitive physical causation between both variables. Specifically, shared external atmospheric drivers can influence both evaporation and precipitation, potentially confounding the identified causal links."*

- Sensitivity of soil-moisture memory and coupling metrics: The soil-moisture memory metric and coupling indices are central to your mechanistic narrative (snow → melt timing → SM → LH → temperature/precipitation). However, these diagnostics can be sensitive to forecast drift and lead-time dependence. Please consider additional clarification and/or sensitivity analyses addressing these dependencies (e.g., lead-time stratification, drift treatment, or related robustness checks).

➔ *We appreciate the reviewer's insightful comment regarding the potential sensitivity of soil-moisture memory (SMM) and coupling metrics to forecast drift and lead-time. To address this, we have added clarification in Section 3.1 and 3.2 regarding how we treated the forecast data to minimize the impact of drift. Specifically, the SMM was calculated using concatenated time series of 1-month forecasts from each prior month's initialization, which helps mitigate the influence of long-term model drift that often intensifies at longer lead times.*

*"In the calculation of the SMM in both seasonal forecast systems, to minimize the impact of systematic forecast drift, the JJA SM time series for each year are constructed by concatenating 1-month lead forecasts for each respective month, specifically June from the 1 May initialization, July from 1 June, and August from 1 July."*

*"The analysis is conducted using MJJA time series of 24-year forecast runs initialized on 1st March for each forecast experiment and ensemble member. The results exhibit a negligible discrepancy with the analysis of the JJA time series (not shown), which accounts for forecast drift and seasonality during the transitional period."*

3. Title revision. Please revise the title for concision and to improve compliance with GMD conventions.

- The current title contains redundant phrasing; please simplify it.

- Please include the relevant model name(s) and version number(s) (e.g., the seasonal forecast system and land model), consistent with GMD guidance for model evaluation papers and related categories.

➔ *We change the title of this paper to "**Implementation of a multi-layer snow scheme in the GloSea6 seasonal forecast system: Impacts on land–atmosphere interactions and climatological biases**".*

4. Code and data availability. Please ensure that materials currently listed as "available upon request" are either deposited in a suitable permanent archive or replaced by a fully reproducible access pathway.

This should include explicit versioning, retrieval scripts/queries, and a clear description of any licensing constraints. If open release is restricted for any component, please describe how confidential or controlled access can be provided to others upon request, consistent with journal policy.

➔ *We have updated the "Data availability" and "Code availability". Timely-filtered ESA CCI SM product previously listed as "available upon request" are being prepared for deposition on Zenodo to provide a fully reproducible access pathway. In contrast, the GloSea5 and G6single retrospective forecasts are not publicly hosted, so that this is informed in the "Data availability".*

5. Figure accessibility and color-vision-deficiency (CVD) robustness. Several figures are visually crowded, with too many lines and markers, which reduces readability. In addition, some key figures rely on color combinations that may be difficult for readers with common forms of color-vision deficiency (notably red–green distinctions in line plots). Because these figures convey central results, please revise them so that the information remains clear without reliance on color cues (e.g., by reducing visual complexity and using line styles, marker shapes, and adequate luminance contrast).

➔ *We have revised Figures 1 and 2 to ensure accessibility for readers with color-vision deficiency (CVD). We replaced red-green color scales with CVD-robust palettes (e.g., blue-red) and used distinct line styles (solid vs. dashed) and marker shapes to distinguish experiments without relying solely on color.*